# Integrating end-to-end learning with deep geometrical potentials for ab initio RNA structure prediction

Yang Li [1,2,8], Chengxin Zhang [2,3,8], Chenjie Feng[2,4,8], Robin Pearce[2,5], P. Lydia Freddolino [2,6] ✉ & Yang Zhang [1,2,5,6,7] ✉

RNAs are fundamental in living cells and perform critical functions determined by their tertiary architectures. However, accurate modeling of 3D RNA structure remains a challenging problem. We present a novel method, DRfold, to predict RNA tertiary structures by simultaneous learning of local frame rotations and geometric restraints from experimentally solved RNA structures, where the learned knowledge is converted into a hybrid energy potential to guide RNA structure assembly. The method significantly outperforms previous approaches by >73.3% in TM-score on a sequence-nonredundant dataset containing recently released structures. Detailed analyses showed that the major contribution to the improvements arise from the deep end-to-end learning supervised with the atom coordinates and the composite energy function integrating complementary information from geometry restraints and end-to-end learning models. The open-source DRfold program with fast training protocol allows large-scale application of high-resolution RNA structure modeling and can be further improved with future expansion of RNA structure databases.

RNA molecules perform a broad range of important cellular functions, ranging from gene transcription, regulation of gene expression and scaffolding, to catalytic activities. The critical functional roles of RNAs make them a new type of drug target. For example, it is estimated that targeting RNAs with small molecules will expand the drug design landscape by more than an order of magnitude compared to traditional protein-targeted drug discovery[1]. Since many biological functions depend on the specific tertiary structures of RNAs, it is imperative to determine the 3D structures of RNAs in order to facilitate RNA-based function annotation and drug discovery. The biophysical experiments capable of resolving RNA structures, e.g., X-ray crystallography, Cryogenic Electron Microscopy (Cryo-EM) and Nuclear

Magnetic Resonance (NMR) Spectroscopy, are unfortunately cost- and labor-intensive. Therefore, fast and accurate computational approaches for sequence-based RNA structure modeling are highly needed.

Traditional RNA structure prediction approaches are often designed to construct 3D RNA structures from homologous modeling approaches and/or physics-based simulations. For example, methods such as ModeRNA[2] and RNABuilder[3] extract structure information from previously solved homologous structural templates. For RNA targets with divergent sequences or novel topologies, their performance is usually unsatisfactory. Another family of methods, typified by RNAComposer[4] and 3dRNA[5], assemble full-length RNA structures from fragments searched from a prebuilt fragment library. Ab initio RNA

[1]Cancer Science Institute of Singapore, National University of Singapore, 117599 Singapore, Singapore. [2]Department of Computational Medicine and Bioinformatics, University of Michigan Medical School, Ann Arbor, MI 48109, USA. [3]Department of Molecular, Cellular, and Developmental Biology, Yale University, New Haven, CT 06511, USA. [4]School of Science, Ningxia Medical University, Yinchuan 750004, China. [5]Department of Computer Science, School of Computing, National University of Singapore, 117417 Singapore, Singapore. [6]Department of Biological Chemistry, University of Michigan Medical School, Ann Arbor, MI 48109, USA. [7]Department of Biochemistry, Yong Loo Lin School of Medicine, National University of Singapore, 117596 Singapore, Singapore. [8]These authors contributed equally: Yang Li, Chengxin Zhang, Chenjie Feng. ✉e-mail: lydsf@umich.edu; zhang@zhanggroup.org

structure prediction methods, such as SimRNA[6], FARFAR2[7], and RNA-BRiQ[8] apply statistical potentials to guide the structure folding simulations. Although utilizing domain expert knowledge with these methods could lead to somewhat better performance in RNA-Puzzles and CASP[9,10], their performance is often suboptimal in automated benchmark test runs, suggesting that automatic prediction of regular RNA structures remains a challenging task to the ab initio simulations.

Deep machine learning has recently demonstrated promising performance in RNA structure feature prediction. For example, SPOT-RNA[11], MXfold2[12] and Ufold[13] utilize convolutional neural networks (CNNs) or recurrent neural networks (RNNs) to improve the accuracy of secondary structure (SS) prediction for RNAs. Analogous to the highly successful efforts in protein contact and distance prediction[14], SPOT-RNA-2D[15] and RNAcontact[16] applied deep residual networks to learn inter-nucleotide distance/contacts from profile covariance. Despite the interest in property learning, few studies have performed full-length modeling of RNA tertiary structures. Recently, a deep equivariant model, ARES[17], was trained to score the conformations from limited data. However, it requires sufficient pre-sampled conformations, which may limit the application.

Inspired by the recent success of deep learning techniques in 3D protein structure prediction[18–20], we proposed a different deep learning pipeline, DRfold, to improve the performance of ab initio RNA structure prediction. Different from the full-atom end-to-end training in proteins[18], we adopted a coarse-grained model of RNA specified by the phosphate P, ribose C4′, and glycosidic N atoms of the nucleobase, for training efficiency given the limited availability of RNA structures. In particular, we added a geometric module which is trained in parallel to assist the end-to-end training, and meanwhile aggregated both end-to-end and geometric potentials to guide subsequent RNA structure reconstruction simulations. We found that the integration of the end-to-end training and deep geometric learning, followed by the gradient-based optimization, generates RNA structure models with accuracy significantly beyond the models solely built on the coarse-grained end-to-end learning or geometry-based structural optimization alone.

## Results

The DRfold pipeline is outlined in Fig. 1. The query sequence along with its SS predictions is first fed into an embedding layer which outputs the sequence and pair representations. The embedded representations are then passed through 48 RNA transformer blocks and are used for end-to-end RNA global-frame training, in terms of nucleotide-wise rotation matrices and translation vectors, which can be used to recover the atomic coordinates of the RNA structure. Meanwhile, the pair representations are also used for RNA inter-nucleotide geometry prediction through a similar but independent set of transformer blocks. Next, the predicted frame vectors and geometric restraints are aggregated into a composite potential for gradient-based RNA structure reconstruction, where the optimized conformation with the lowest energy is selected as the output model. Finally, the coarse-grained models are submitted to a two-step molecular dynamics-based procedure for atomic-level structure reconstruction and refinement.

DRfold was tested on 40 non-redundant RNA structures with lengths from 14 to 392 nucleotides, which were collected from sequence cluster centers (sequence identity cutoff of 90%) of solved structures deposited in or after the year 2021 in the PDB[21]. Structures without any valid base pairs were not included. All test sequences, as well as their cluster members, were excluded from the training dataset, which contains 3864 unique RNAs extracted from the PDB that were deposited before the year 2021. Thus, a filter based on both sequence identity and time stamp implements a stringent separation between the training and testing datasets of DRfold, both of which are available for download at https://zhanggroup.org/DRfold. The length range of the test RNAs [14, 392 NTs] was slightly larger than but generally consistent with the crop-size range of the RNAs that were used for

training the DRfold models (i.e., <200 NTs) due to the GPU memory limitations.

## DRfold outperforms previous RNA structure predictors

To benchmark the performance of DRfold with previous approaches, two representative fragment assembly methods, RNAComposer[4] and 3dRNA[5], and three representative ab initio RNA structure prediction methods, RNA-BRiQ[8], SimRNA[6], and FARFAR2[7], were considered as control methods, where a brief introduction to the configurations of these methods is given in Text S1. Figure 2A compares the root mean squared deviation (RMSD) of the models generated by DRfold and the control methods relative to the target structures, where the coordinates of the P atoms were used for topology evaluation. The average RMSD value obtained by our method (14.45 Å) was significantly lower than those obtained by 3dRNA (20.54 Å), FARFAR2 (22.48 Å), RNA-Composer (20.80 Å), BRiQ (22.88 Å), and SimRNA (23.88 Å), where the corresponding $P$-values obtained by two-tailed Student's t-tests were 7.35E−05, 3.72E−07, 1.90E−04, 3.34E−07, and 6.14E−07, respectively. The median RMSD of DRfold was 9.38 Å, compared to the lowest median RMSD of 19.04 Å obtained by the control methods (RNA-Composer). Among the 40 test targets, 6 (or 2) targets were found to be successfully folded by DRfold at a high accuracy with RMSDs <2.5 Å, as evaluated by the P-atom (or full-atom) RMSD. In Fig. 2B, we further list the accumulative fraction of cases with RMSD values below thresholds ranging from 2.5 Å to 15.0 Å, where DRfold generated significantly more cases than the control methods across all RMSD cut-offs. For example, 47.5% of the DRfold models had an RMSD less than 7.5 Å, which is more than twice the fraction (20.0%) obtained by the best-performing third-party method, 3dRNA.

Since a local error could cause a high RMSD, the RMSD value may not be ideal for assessing the quality of the RNA models at the high RMSD range. In Fig. 2C, we further list the results for the TM-score, an index that is more sensitive to the global fold of the RNA structures[22]. Here, TM-score ranges from (0,1] with a higher value indicating a closer structural similarity, where a TM-score above 0.45 indicates a correct fold for RNA structures independent of the sequence length. As shown in Fig. 2C, the average TM-score of the DRfold models (0.435) was 73.3% higher than the average TM-score of 0.251 obtained by the second-best method, 3dRNA, with a $P$-value of 5.79E−07. Furthermore, 45% (=18/40) of the DRfold models had correct folds with TM-scores >0.45, while the second-best method only achieved a success rate of 12.5%. The ability of DRfold to obtain very high-quality overall models for a substantial fraction of targets is apparent in the large upper shoulder in the distribution of TM-scores shown in Fig. 2C.

In Fig. 3, we present a detailed head-to-head comparison of TM-score and RMSD between DRfold and the control methods, where a pronounced advantage of DRfold over the control methods was observed in all boxes. For example, the fraction of the test targets for which DRfold achieved a lower RMSD than the control methods was 80.0% (to 3dRNA), 82.5% (FARFAR2), 72.5% (RNAComposer), 75.0% (RNA-BRiQ), and 80.0% (SimRNA), respectively, as shown in Fig. 3A−E. The superiority of DRfold was more robust when evaluated by TM-score in Fig. 3F−J, as the maximum absolute difference in TM-score was only 0.039 for those targets where any of the control methods performed better. In contrast, for those targets where DRfold had a higher TM-score, the absolute difference was up to 0.734. Such observation suggests again that DRfold can consistently predict better global conformations compared to the classic RNA folding methods.

To investigate the possible impact of sequence homology cutoffs on the accuracy of the DRfold models, an exhaustive test considering various sequence identity cut-offs between the training and test sets was conducted. Following conventional criteria used in previous studies for RNA structure/SS prediction using deep learning[11,23–25], additional datasets were constructed by excluding targets with sequence identities greater than multiple thresholds (i.e., 80%, 70%, and 60%) to

the DRfold training dataset; this resulted in 32, 23, and 10 sequence-nonredundant RNA structures, respectively. The results show that the performance of DRfold correlates with the sequence cut-offs for the test sets, where the average TM-scores of the selected test targets gradually decreased from 0.435 to 0.309 as the maximum sequence identity cut-off decreased from 90% to 60% (Fig. 2D). These data suggest that the deep learning-based predictions are more reliable when trained on similar sequences. Nevertheless, the average TM-score for DRfold consistently exceeded that of the best control methods by at least 33.9% across all thresholds.

The advantage of DRfold is consistent with our expectations, as existing automatic RNA structure prediction methods mostly utilize basic empirical and statistical potentials of the form $P(structure|sequence)$. Given the limited number of parameters in their force fields, the global sequence conditions cannot be extensively accounted for and the generic potential forms (e.g., distance or angles) do not precisely determine the complex topology of the RNA structures. In contrast, the extensive weighting parameter settings

embedded in the transformer module used by DRfold allow for access to the global sequence information. In addition, the end-to-end loss function (see "Methods") can further ensure the high correspondence of the deep learning predictions with the correct overall conformations.

In Figure S1, we list the scatter plot of TM-score versus the length of the test RNAs, where a weak correlation (Pearson Correlation Coefficient, PCC = −0.20) can be observed, indicating that the performance of DRfold is overall weakly dependent on the RNA length. It is notable that for those targets with lengths >200 NTs, the TM-scores obtained by DRfold are lower overall than those obtained for smaller targets <200 NTs. One reason for the suboptimal performance for large-size RNAs is probably that a maximum RNA length cutoff was set to 200 NTs when we trained the models in DRfold due to the limited GPU memory (with a single Nvidia A40 GPU with 32 GB memory), and therefore the interaction patterns for extremely distant (>200) nucleotide pairs may not be sufficiently learned. Developing length-insensitive variants of attention networks by utilizing more

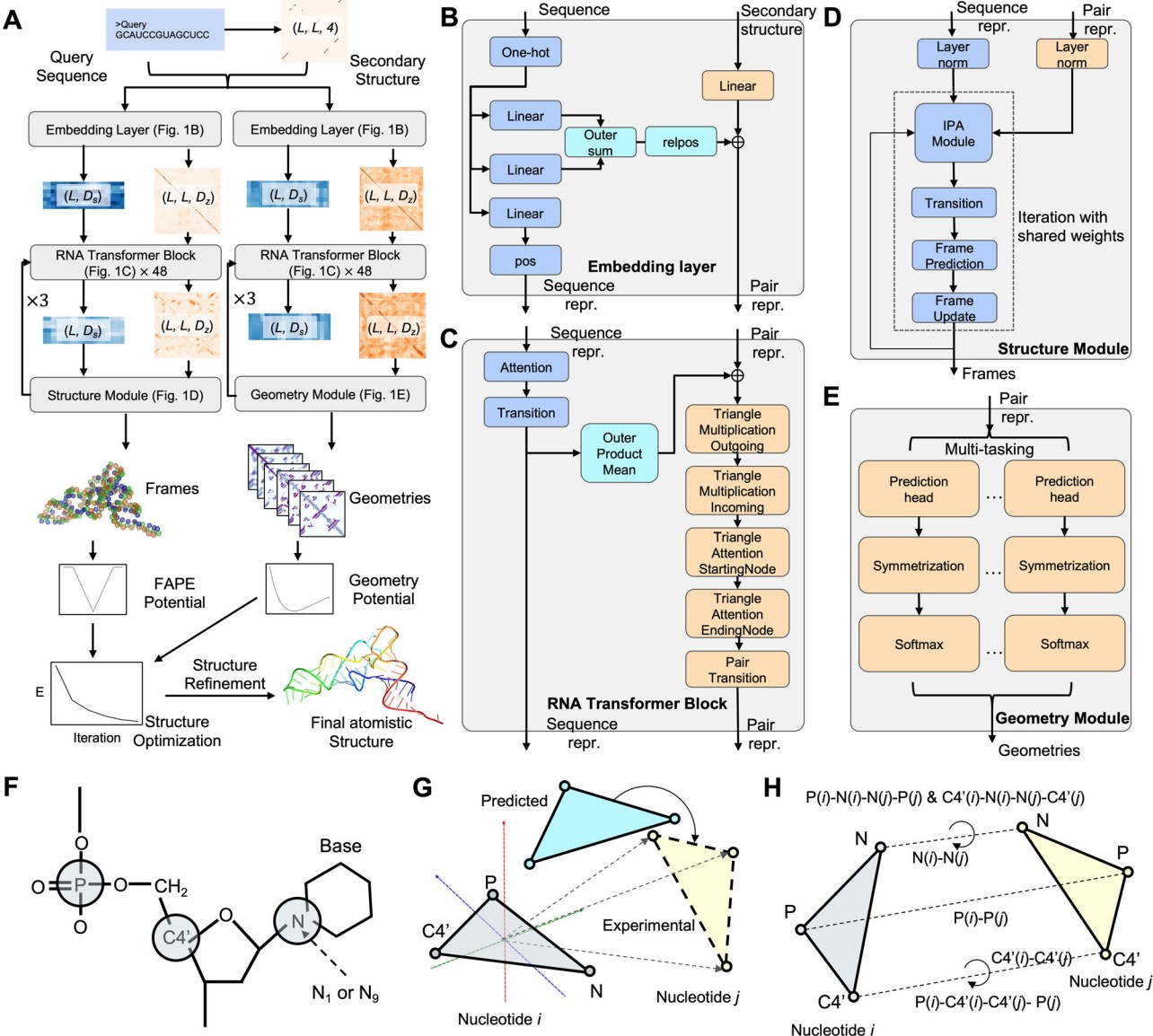

**Fig. 1 | The pipeline of DRfold for deep learning-based RNA structure prediction by combining end-to-end model and geometry potentials. A** DRfold pipeline for sequence-based RNA structure prediction, where $D_s$ and $D_z$ are hidden dimension sizes of sequence and pair features, respectively, and $L$ is the length of the query sequence. **B**–**E** Details of embedding layer, RNA transformer block, and structural and geometry modules, respectively. **F** Reduced representation of nucleotide residues by a 3-bead model (C4', P, glycosidic N) in DRfold. **G** Illustration of the frame aligned point error (FAPE). **H** Prediction terms of inter-nucleotide geometry.

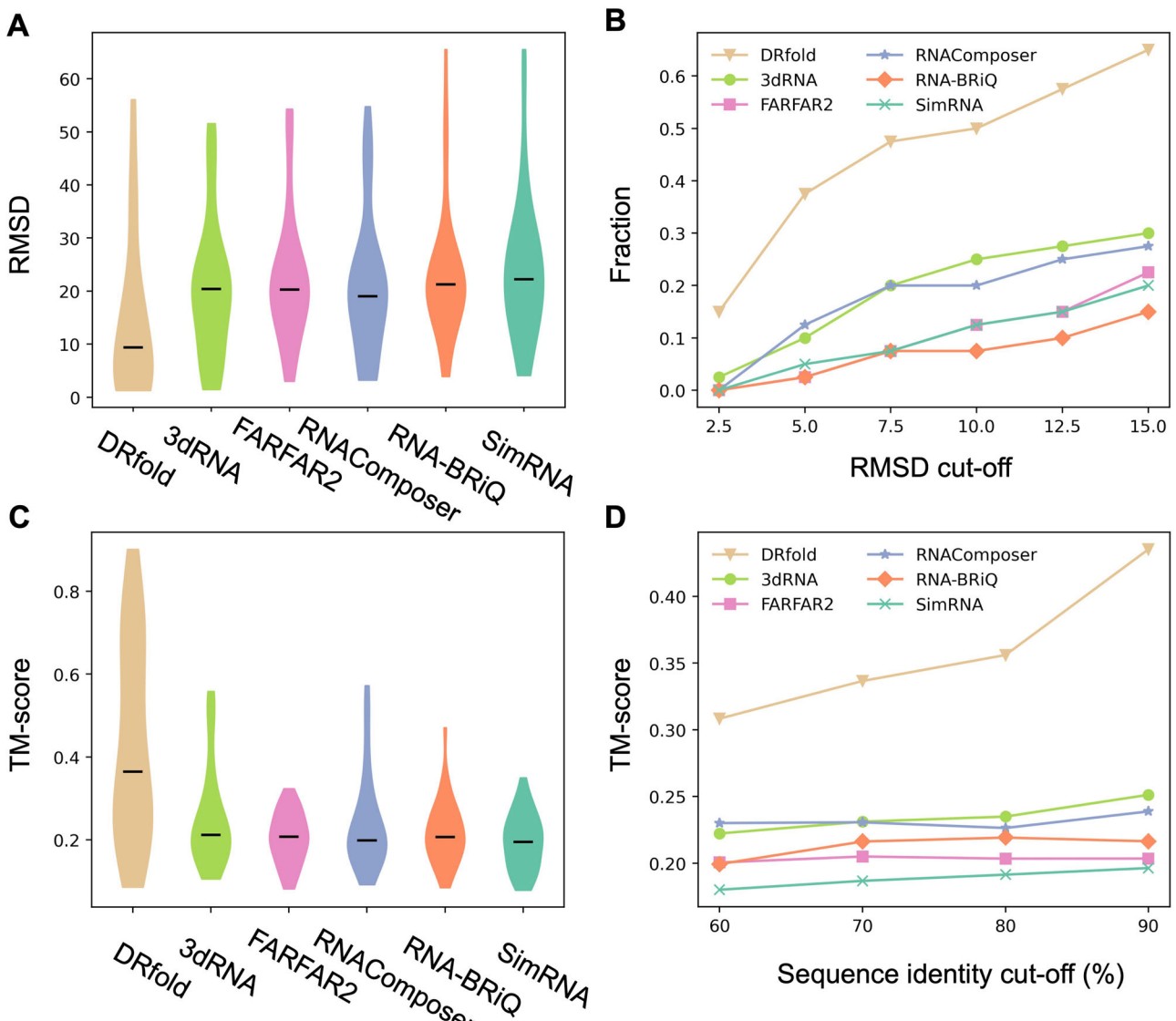

**Fig. 2 | The comparison of DRfold with the control methods. A** Distribution of RMSD (Å) of the predicted models to the target structure. The central mark indicates the median. **B** Fraction of the test RNAs achieving successful structure prediction at different RMSD cut-offs. **C** TM-score distribution of different methods.

The central mark indicates the median. **D** The average TM-score by different methods versus the sequence identity cut-off between the test and DRfold training datasets.

comprehensive RNA dataset and larger computing resources should help DRfold to learn the longer-range inter-residue interactions and therefore enhance its ability to fold large-sized RNA structures.

The hydrogen-bonding interactions between conjugated nucleotides are critical to stabilize the tertiary structures and functions of RNAs. It is therefore useful to investigate whether and how DRfold can recover these SS patterns. In Table S1, we summarize the base interaction network fidelity (INF)[26,27] and deformation index (DI)[28] scores of the models generated by DRfold and the control methods, which were calculated using the RNA-Puzzles toolkit[29]. Here, the INF is defined as Matthews Correlation Coefficient (MCC) between the interactions of the reference structure and that of the predicted structure, and it was split into four categories according to the interaction types, including Watson-Crick (INF_wc), non-Watson-Crick (INF_nwc), stacking (INF_stack), and overall interactions (INF_all). The DI is defined as the RMSD between two optimally aligned 3D structures divided by the base INF and can reflect the overall features (encoded by the RMSD) calibrated by the quality of the reproduced interaction network (encoded by the INF value). Although DRfold does not employ specific base-pairing

related potentials, it outperforms other methods across each evaluation index, suggesting that the relative frame positions in the frame aligned point error (FAPE) and geometrical potentials may have implicitly helped DRfold to recover the base-pairing patterns of the structure models (see "Methods"). In the lower panel of Table S1, we also list the performance comparisons on the targets with a sequence identity cut-off of 80% to the DRfold training set, where DRfold still shows an advantage compared to the automatic control methods. It is noted that despite the overall advantage, the success rate of non-canonical base-pairing prediction (INF_nwc) by DRfold was still low. A more detailed learning model at the atomic level trained on the datasets with enhanced non-canonical pairing samples might help improve the accuracy for INF_nwc.

In Table S2, we further list the mean of circular quantity (MCQ)[30] and Handedness scores of the DRfold models compared to the five control methods. Here, the MCQ score measures the dissimilarity between two structures in torsion angle space using full-atom representations[30], assuming the standard bond lengths and bond angles are constant values. In addition, the Handedness score is

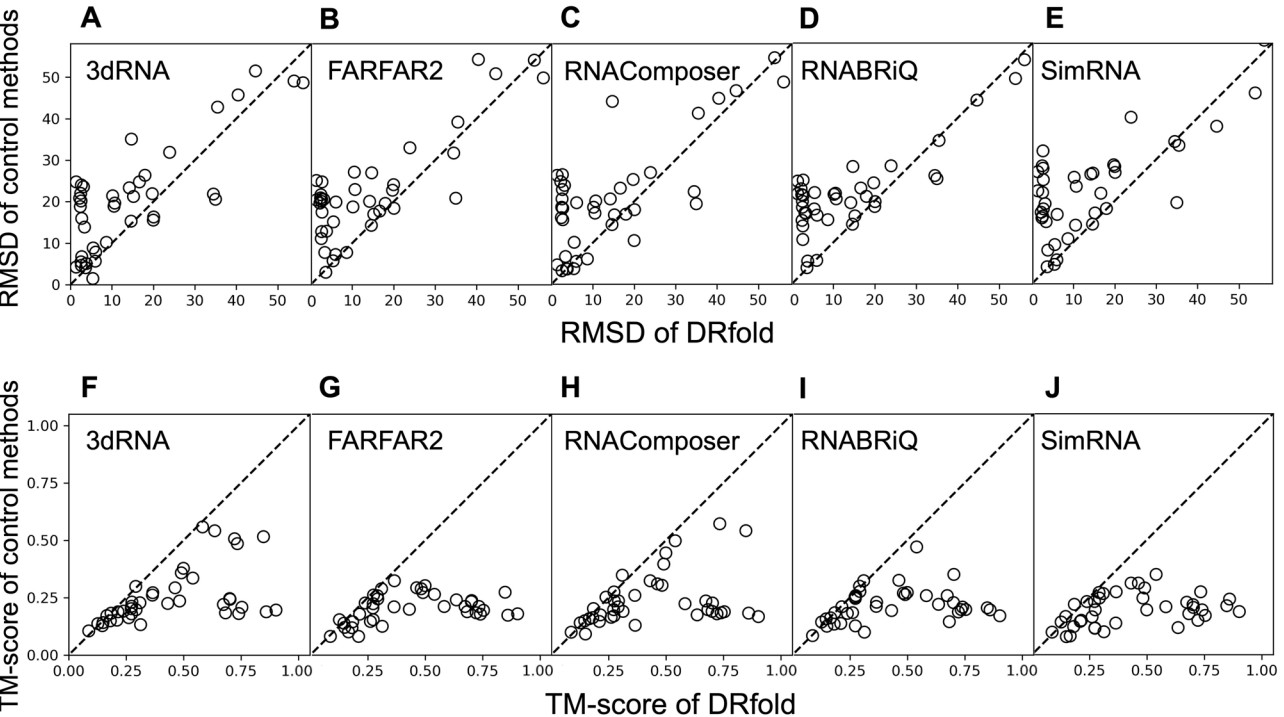

**Fig. 3 | Head-to-head comparisons between DRfold and the control methods on the 40 test RNA structures.** A–E RMSD (Å). F–J TM-score.

introduced to evaluate the correctness of the chirality of the RNA helices, which computes the fraction rate of non-loop residues in the predicted models that have closer C4′ torsion angles to the targets than to the mirror images of the target structures. The results show that while DRfold does not outperform the control methods in MCQ, likely due to the coarse-grained representation of RNA structures during the model training procedure, DRfold excels in Handedness score over all control methods. This performance could be attributed to the use of mirrored-sensitive potentials by DRfold, including the reflection transformation-sensitive FAPE potential[18] and the use of long-range dihedral angles in the geometry potential, which are capable of distinguishing the desired structures from their mirror images.

Out of the 40 test targets, 15 contain pseudoknots as assigned by DSSR[31]. While 3dRNA and RNAComposer cannot detect any of the pseudoknots for the 15 targets, SimRNA and FARFAR2 produce 6 and 2 structures that contain pseudoknots, respectively; however, none of the detected pseudoknots by SimRNA and FARFAR2 have correct correspondence to those in the native structures. DRfold predicts two structures that have pseudoknots assigned, as shown in Figure S2. It can be observed that DRfold can correctly recover the pseudoknots in both cases, highlighting the ability of the DRfold networks to learn complex inter-nucleotide interaction patterns.

**End-to-end models provide complementary information to geometric restraints for RNA structure modeling**
The core of the DRfold pipeline is the introduction of two types of complementary potentials, i.e., the FAPE potential and the geometry potentials, from two separate transformer networks. The former models directly predict the rotation matrices and the translation vectors for the frames representing each nucleotide, forming an end-to-end learning strategy for RNA structure prediction. In DRfold, 6 independent end-to-end models were trained with different parameter initializations, where Table S3 lists the average TM-score of those models on the test set. Without any post-processing, the individual end-to-end models already outperform all control methods significantly. For example, the lowest average TM-score (0.393) obtained among the 6 end-to-end models was 57.2% higher than that of the best

control method 3dRNA (0.250). After applying an optimization procedure which is an ensemble of the 6 conformations, the average TM-score rose to 0.417.

To further examine the importance of the end-to-end potential to DRfold, we plot in Fig. 4A a TM-score comparison of the models generated by the full-version of DRfold with those generated without the FAPE potential in DRfold, where the latter indicates that the structures were only optimized by the geometry potentials. Without considering the atomic-level refinement, the average TM-score of raw DRfold dropped from 0.439 to 0.413, with a P-value of 2.7E−02 as determined by a two-tailed Student's t-test, indicating that the performance loss is statistically significant. In Fig. 4C−E we present one example from the sgRNA (PDB ID: 7OX9 Chain A) in the Cas9 endonuclease. The model built using the geometry potential has a reasonable fold but with significant local errors mainly in the 5′- and 3′-terminal regions and the central loop (26−41 NTs), which resulted in an overall TM-score = 0.369 and RMSD = 6.52 Å. As shown in Fig. 4E, the end-to-end structural models have variable quality in the 5′- and 3′-terminal regions with consistently lower errors in the loop region. A consensus-based optimization of both end-to-end and geometric potentials resulted in a significantly improved model with a TM-score = 0.749 and RMSD = 2.00 Å (see Fig. 4D and the bottom of Fig. 4E).

The geometry potential in DRfold adopts a composite set of terms representing inter-nucleotide geometry, including distances and torsion angles. To examine the impact of such potentials to DRfold, we compare the TM-scores of DRfold on the 40 test RNAs with and without the geometry potentials in Fig. 4B. It was observed that including the geometry potentials on top of the end-to-end potential brings small but consistent improvements in TM-score (with P-value = 2.1E−07). In Figure S3, we list the TM-score improvements by the geometry potentials over the RNA length, which shows that the incorporation of the additional long-range inter-nucleotide geometric restraint potential can consistently improve the folding performance for RNA with various sizes.

Overall, these results demonstrate that, although they start from the same set of sequence and SS embedding matrices in the network, the independent training of the end-to-end and geometric potentials

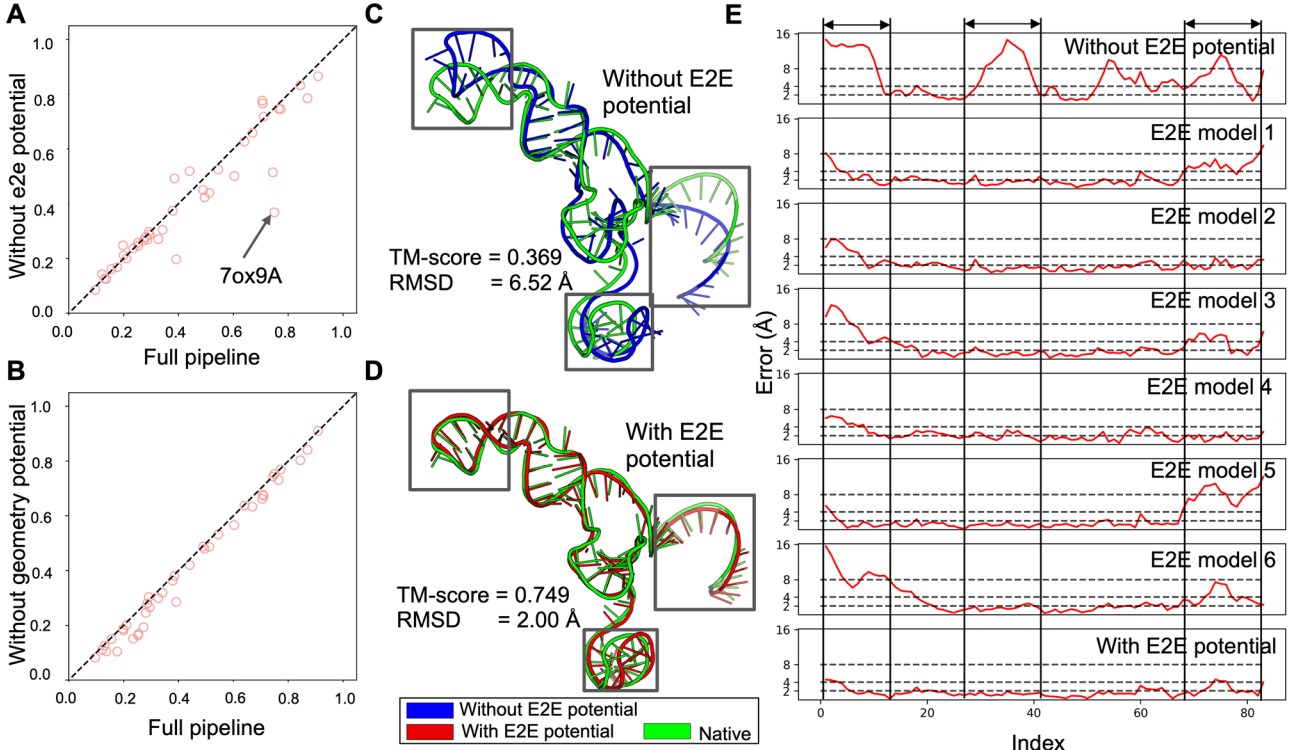

**Fig. 4 | Performance comparisons between the full DRfold pipeline and those without component potentials.** TM-score comparison of full versus ablated pipelines **A** without end-to-end potential and **B** without geometry potential. **C** Structure superposition between target structure (green) and the structure predicted without end-to-end potential (blue) for target 7OX9A. **D** Structure superposition between target structure (green) and the predicted structure with the full DRfold pipeline (red) for target 7OX9A. **E** Residue-wise errors of the structure predicted without end-to-end potential, the structures predicted by 6 end-to-end models, and the structure predicted by the full DRfold pipeline for target 7OX9A. The residue-wise errors were computed from the superpositions produced by TM-score.

learned structural features complementary to each other and collectively improved the overall quality of the DRfold models.

**Secondary structure prediction facilitates feature learning and model construction**
Unlike the strategy used in AlphaFold2[18] which feeds the embedding module directly with the query sequence, DRfold uses a consensus of two SS predictors by RNAfold[32] and PETfold[33] to extract 2D features for the additional pair embedding. To examine the impact of the SS predictions, Fig. 5A shows a head-to-head comparison of TM-score for the full DRfold pipeline vs. an ablated pipeline in which the predicted SS input is omitted. There is a significant performance drop, i.e., from 0.439 to 0.295 in TM-score, without the inclusion of secondary structure information.

As a complementary test to determine the amount of information on tertiary structure that can be obtained from secondary structure alone, we performed an additional experiment in which we replaced all input residues with an "N" (which represents the "unknown" residue type) and only fed DRfold with the predicted secondary structure features, mimicking a tertiary structure recovery task from SS prediction alone. Unsurprisingly, the average TM-score significantly dropped from 0.439 to 0.268. Nevertheless, the TM-score value was still considerably higher than that by the best third-party program, 3dRNA, based on statistical models (0.250). Among the 40 test targets, there were 5 targets that had correct folds with TM-scores >0.45 for the sequence-free DRfold modeling.

In Fig. 5B, C, we show one example in which DRfold successfully recovered the overall topology only based on its predicted secondary structure. This is a tRNA (PDB ID: 7MRL Chain A), which has a clover-like structure. Based on the predicted SS only (Fig. 5B), DRfold

constructed a model of correct fold with a TM-score = 0.564 and RMSD = 3.19 Å (Fig. 5C). Nevertheless, we noticed that many of the base-pairs were not correctly formed, due to the lack of nucleotide sequence information. After the inclusion of the sequence information, the RNA model by the full DRfold pipeline had much-improved base-pairing quality with an overall TM-score=0.765 and RMSD = 2.22 Å (Fig. 5D), demonstrating the impact of sequence-specific base-pairing on the RNA structure modeling.

Overall, these results demonstrate the significant importance of the SS embedding feature to the DRfold performance, although the pipeline starts from the nucleotide sequence only. In principle, an ideal deep learning model should be able to learn the secondary structure directly from sequence. Previous studies[11–13] have shown success in learning RNA secondary structure directly from sequence. However, given the limitation of available RNA structure data, relevant input structural feature, containing auxiliary information related to the RNA topology such as SS, should be greatly beneficial to facilitate the neural networks to improve the learning efficiency and effectiveness in RNA tertiary structure prediction.

Given the special role of SS in RNA tertiary structure prediction, we further tested DRfold with two other types of SS inputs from either SPOT-RNA predictions or extracted from the target structures. Table S4 compares the average TM-score and RMSD of raw DRfold predictions under three conditions with different SS features. Compared to the default settings, the SPOT-RNA-based SS feature provides comparable overall performance, with a slightly lower average TM-score but lower average RMSD, despite DRfold not being trained with this SS model. As expected, the SS features extracted from the experimental structures (Ground-Truth in Table S4) yield the best performance compared to other SS features. However, the superiority

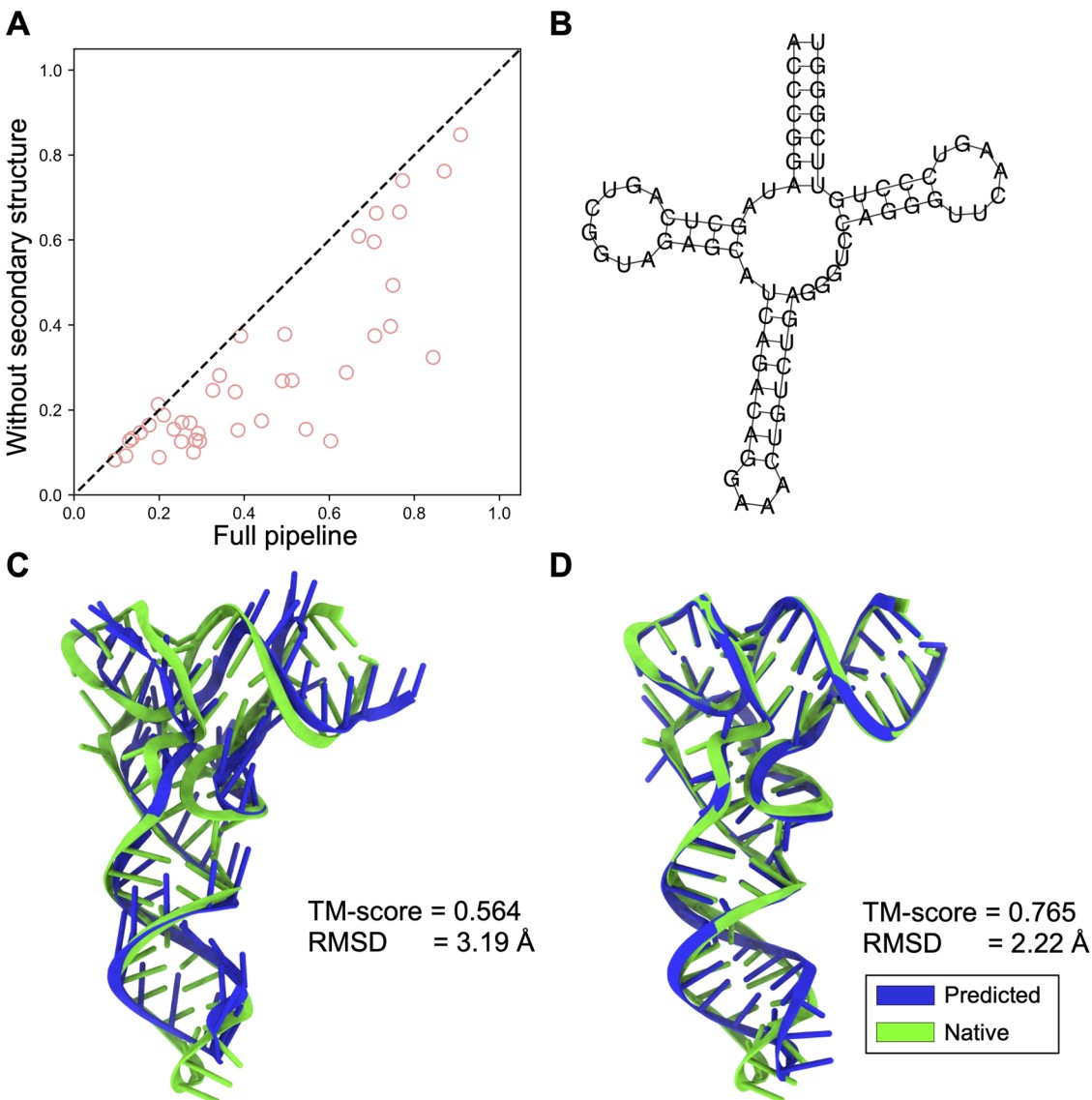

**Fig. 5 | Secondary structure feature improves performance. A** TM-score comparisons between the full DRfold pipeline and an ablated model without predicted secondary structure information. **B** The predicted secondary structure for target 7MRLA. **C** Superposition of the predicted structure (blue) by DRfold using only the secondary structure with the experimental structure (green) for target 7MRLA. **D** Superposition of the predicted structure (blue) by the full version DRfold with the experimental structure (green) for target 7MRLA.

is somewhat limited, as the average TM-score using ground-truth SS features is only 0.91% and 2.78% higher than that using two predicted SS features, respectively. One possible reason for the modest improvement from using the native SS assignment is that the DRfold was trained based on predicted SS information with noises and therefore the modeling weights associated with the SS component may not be strong enough to count for the true SS assignments. To examine this possibility, we made a further test on the end-to-end component of the DRfold pipeline with both default and retrained parameters when using new SS assignments. As shown in Table S5, although the use of native SS still results in significantly better performance than the predicted SSs, there is no appreciable difference on the average TM-score and RMSD between the models using default and retrained parameters, suggesting that the default models are robust and no retraining is needed when using different SS assignments.

Overall, the data show that the current DRfold model is not overly sensitive to the SS feature, as long as SS predictions are of reasonable accuracy, where all tested SS models have an MCC above 0.670. On the

other hand, the current 2D SS features from traditional Watson-Crick base-pairing predictions are relatively simple, while input matrices with more specific SS information, such as pseudoknots or inter-helical interactions, and richer descriptors, such as distances or torsion angles, may be helpful to further leverage the local SS features for more accurate RNA structure prediction.

### Structure refinement improves the physical realism of DRfold models

The learning models and potentials utilized in DRfold are coarse-grained built on the P, C4', and N atoms (Fig. 1F). To enhance the validity and biological usefulness of the predicted models, a two-step refinement process was designed to obtain atomic-level, fine-grained models. Table S6 presents a comparison of the RNA structure validity parameters through different stages of structural refinement. The raw DRfold models exhibit a high clash score (224.15), resulting in a high MolProbity score[34] (3.95). Arena, an in-house program (see "Methods"), was employed to quickly reconstruct full-atom RNA structures and rectify incorrect base conformations on top of raw coarse-grained

models, termed as Step 1. Subsequently, the clash and MolProbity scores decreased to 82.79 and 3.42, respectively. Step 2 involved the use of OpenMM[35] to further refine the structures through AMBER-guided[36] all-atom molecular dynamics (MD) minimization, leading to a significant decrease in the clash (18.57) and MolProbity scores (2.42). Accordingly, the bond length and torsion angle violations, assessed by the root mean square (RMS) deviations from their restrained ideal values, were reduced from 0.06 and 10.99 to 0.03 and 4.21, respectively, after the two-step refinement procedure.

The validity parameters from experimental structures in the PDB are also listed as a reference at the bottom of Table S6. Prior to refinement, the MolProbity score of DRfold was 63.2% higher than that of native structures. Following refinement, the MolProbity score difference between the DRfold models and experimental structures decreased to 16.9%. Moreover, the clash score, bond length and torsion angle variations of the final DRfold models all became much closer to those of the experimental structures after the atomic level structure refinement. As a tradeoff, the global model quality assessed by TM-score suffered only a very minor decrease from 0.439 to 0.435.

### DRfold produces competitive predictions to cutting-edge deep learning methods

Most recently, several deep learning models have been proposed for RNA structure prediction. Table S7 summarized the results of DRfold on the 40 test RNAs compared to five publicly released deep learning methods: ARES[17], DeepFoldRNA[24], RhoFold[23], RoseTTAFoldRNA[37], and trRosettaRNA[25]. Depending on the input features that the models were trained on, these methods can be classified into either single sequence-based or multiple sequence alignment (MSA)-based approaches. Although MSA-based methods can benefit from co-evolution information derived from MSAs and therefore often achieve better performance on overall structure prediction[18,38], training on single query sequences has advantages in terms of the speed and flexibility of modeling as the procedure does not rely on the construction of MSAs, which can often be tedious and complicated[39].

The data in Table S7 show that DRfold significantly outperforms other single sequence-based approaches, including the previous control methods and FARAFA2/ARES, with p-values as determined by two-tailed Student's t-tests ≤1.05E−06. Although DRfold was trained on single sequences, it achieved comparable performance with most of the MSA-based approaches. For instance, DRfold had a higher TM-score (0.435) than RhoFold (0.420) and RoseTTAFoldRNA (0.428), but lower than DeepFoldRNA (0.485) and trRosettaRNA (0.474); the differences between them were not statistically significant except for DeepFoldRNA which had a p-value = 1.66E−02 against DRfold.

It should be noted that a non-redundancy filter between the training and test datasets was stringently implemented for DRfold, but the same was not implemented between the training sets of the control methods and the 40 test targets used in this study. For example, we found that 26 targets in our test dataset had a sequence identity above 90% to the trRosettaRNA training dataset. If we exclude these 26 targets, the average TM-score of trRosettaRNA will be reduced to 0.422, which is considerably lower than that of DRfold (0.476). This result again suggests that the performance of most deep learning RNA structure prediction methods may depend on the sequence similarities between the target and training sequences.

From a methodological perspective, the aforementioned deep learning methods can be classified as end-to-end approaches (RhoFold and RoseTTAFoldRNA) or geometry-based approaches (trRosettaRNA and DeepFoldRNA). DRfold, however, combines both approaches through potential integration. The integration provides DRfold with flexibility in its pipeline expansion. For instance, we can use Deep-FoldRNA's geometry predictions to construct hybrid geometry potentials and replace the default geometry potentials in DRfold. For this, we combined the end-to-end potentials (from DRfold) and geometry potentials (constructed from DeepFoldRNA predictions) into a new hybrid potential and use it to guide the subsequent structure optimization, while keeping other part of the DRfold procedure unchanged. In such setup, because DeepFoldRNA focuses on training precise inter-nucleotide geometry terms (e.g., distances and orientations) by leveraging co-evolution from MSA and unlabeled RNA sequences, it can provide extra and sometimes more accurate spatial restraints than the DRfold geometry restraints; thus, a hybrid potential with complementary and more accurate restraints helps better guide the structural assembly and refinement process of DRfold pipeline.

As shown in Table S7, without any further parameter optimization, the hybrid pipeline (DRfold/DeepFoldRNA) achieves an average TM-score of 0.501, which is 3.3% higher than the best individual program, DeepFoldRNA (0.485), for the 40 test targets. Although the major goal of this study is to develop new standalone pipeline for independent RNA structure modeling, the experiment does show the methodological flexibility of DRfold to combine other methods for further improving its ability for higher-accuracy RNA structure prediction.

### Blind RNA structure prediction in CASP15

An early version of the automated DRfold program participated in the recent community-wide CASP15 experiment for RNA structure prediction[40] with Group ID 'rDP'. Although there were only 12 test targets[41], this gave an opportunity to objectively assess DRfold relative to the state of the art of the field. In Tables S8 and S9, we list the cumulative Z-scores for all groups in terms of RMSD and TM-score of the first predicted models, respectively. Following the convention of CASP, the Z-scores were calculated using the following procedure: (1) for a given target raw Z-scores were calculated as the difference between the raw score and the mean in the unit of standard derivation for the first models of different groups; (2) remove the outlier models with raw Z-scores below the tolerance threshold (set to −2.0); (3) recalculate Z-scores on the reduced model set; (4) assign Z-scores below the penalty threshold (either −2.0 or 0.0) to the value of this threshold. As shown in Table S8, using RMSD Z-score (calculated by negative of RMSD values), DRfold ranks 5th and 6th with penalty thresholds of −2.0 and 0.0, respectively. When using TM-score, the ranking becomes 6th and 9th, respectively (Table S9).

In Table S10, we further list the comparisons of average RMSD and TM-score of the groups that have submissions for all CASP15 targets, where DRfold ranks 4th and 9th on RMSD and TM-score, respectively, which are largely consistent with its ranking on Z-scores. Meanwhile, we found that the average values of TM-score/RMSD on the CASP15 targets (=0.288/21.60 Å, or =0.302/20.34 Å after excluding the super-long 720 NTs target R1138) are largely consistent with the benchmark test results on the targets with a sequence identity cutoff of 60% to the training dataset (=0.309/24.27 Å). Considering that all CASP15 targets also have a sequence identity below 60% to the DRfold training dataset, this result demonstrates the robustness of the benchmark test and generalization ability of the DRfold on modeling different RNA structures.

There is an obvious performance gap between DRfold and the top 4 methods (AIchemy_RNA2, Chen, RNApolis, and GeneSilico), which fold RNAs guided by highly specialized human-expertise in the field of RNA structure. Our method, in contrast, only requires single sequence information and is fully automatic. Nevertheless, we found that the performance of DRfold is comparable with that of other top methods which require additional information sources such as templates (e.g., CoMMiT-server), MSAs (e.g., AIchemy_RNA, Yang-Server, and Ultra-Fold) or pretrained nucleotide sequence models (e.g., AIchemy_RNA). Considering that the only input of DRfold is the RNA sequence, the reason for such competitive performance should be attributed to the network of DRfold that learns complementary potentials to improve RNA folding. Meanwhile, the excellent performance of other top-

ranked methods demonstrated potential to further improve DRfold with the integration of those additional information sources.

## Discussion

We developed a novel method, DRfold, for ab initio RNA structure prediction through single sequence-based deep learning models. The approach is able to learn the coarse-grained RNA structures directly from sequences in an end-to-end fashion, using cutting-edge self-attention transformer networks. The predicted conformations are further optimized by integrating a separately trained deep-geometric potential through gradient-descent-based simulations. DRfold was tested on a nonredundant set of RNA structures, which were separated from the training RNAs with a strict control of release time and sequence identity cutoff, where the results showed significant advantage over existing approaches built on statistical model and fragment assembly simulations. Despite the simplified training on single sequences, DRfold generated models with competitive accuracy to the sophisticated approaches trained on MSAs, where a simple hybrid of DRfold with another MSA-based approach (DeepFoldRNA) outperformed all state-of-the-art deep learning approaches on the benchmark tests.

The success of DRfold mainly arises from the deep learning-based potentials, which, to our knowledge, have rarely been introduced in existing RNA structure prediction pipelines. The end-to-end models in DRfold have proven to be highly effective in predicting the frames of residue positions that can recover the atomic model of RNA structures through trained rotation and translation matrices. With the integration of geometric restraints, the hybrid potentials can further improve the accuracy for structural models through atomic-level optimization. Moreover, the predicted secondary structure features from physical-based folding programs were found beneficial to facilitate the network learning and help generate more accurate base-pairing and local structural packing for the RNA models.

Despite the success demonstrated here, we found that the overall performance of structure prediction for RNA is still limited, compared to that for proteins (for example, AlphaFold2[18]). This may be partly due to the lack of a sufficient number of RNA structures that are needed to train the high number of parameters in the end-to-end networks. This is especially true for the CASP15 targets[41], where DRfold performed better on the natural RNAs than the synthetic RNAs that may bear different folding pattern from the natural RNAs on which DRfold was trained. The second limitation of DRfold comes from the present GPU memory resource which allows model training only on small RNAs <200 NTs and, together with the limitation of the training structural repositories, made the current models fall short on large-size RNA modeling as demonstrated in both the benchmark and CASP15 experiments. Finally, to facilitate the modeling of hard RNAs without many homologous sequences, DRfold was trained only on single sequences, while the inclusion of multiple sequence alignments[38], structure templates, and RNA physical knowledge (e.g., atomic patterns of pseudoknots and base pairs) might help to improve the RNA structure prediction through the aggregation of more extensive evolutionary features or the incorporation of general knowledge-based potentials in the simulations. Nevertheless, as a proof of concept, the release of the open-source DRfold program provides a flexible and useful platform to the community for efficient deep learning-based RNA structure prediction whose accuracy will continue to improve with the progress of new machine learning techniques and RNA structure and sequence databases.

## Methods

DRfold is a deep machine learning-based approach to ab initio RNA structure prediction. It consists of three steps of end-to-end frame and geometric potential training, deep-potential-guided 3D structure assembly simulations, and full-atom structure reconstruction and refinement, where the flowchart of the pipeline is shown in Fig. 1.

### Feature preparation and embedding

The only required input of DRfold is the nucleotide sequence, which is represented by a 5-D one-hot encoded vector, including 4 types of nucleotides ('A', 'U', 'G', 'C') and an unknown state ('N') representing modified or degenerate nucleotides. The last state is added to avoid possible training noise brought by the uncertainty of the nucleotides. Therefore, the DRfold model can model RNAs with 5 states ('A', 'U', 'G', 'C', 'N'). At the last stage of atomic structure reconstruction and refinement, however, the residue 'N' will be mutated to either the smallest base 'U' (for unpaired residues) or the conjugated base (for paired residues) for full-length RNA structure prediction.

Based on the query sequence, the SS is predicted by two complementary methods: RNAfold[32] and PETfold[33], which are concatenated in the network. Here, consistent with the requirement of DRfold, both RNAfold and PETfold are configured with sequence input only. The predicted SS is in the form of a matrix, where the entry is set to 1 if the corresponding residue pair forms a base pair. We also include the SS probability map predicted by the considered methods, which brings in another 2 channels for the pair input. Thus, given a sequence of length $L$, the sequence feature (1-D) and the pairwise feature (2-D) have the shapes $L \times 5$ and $L \times L \times 4$, respectively, where '4' is from 4 SS channels (Fig. 1A).

The query sequence feature $s_F \in R^{L \times 5}$ and the pair feature $z_F \in R^{L \times L \times 4}$ act as the input of the embedding layer. Here, the embedding layer is a neural network module that transforms the input features ($s_F$ and $z_F$) into learned representations (sequence representation $s$ and pair representation $z$) through a set of linear layers. After the embedding layer, each residue and residue-residue pair will be represented by a 64-D vector. This layer of embedding is important for generating hidden representations ($s$ and $z$) as the input for the subsequent RNA transformer networks. Specifically, the sequence feature $s_F$ will be projected to the desired dimension ($D_s = 64$) by a linear layer. Another two linear projections of $s_F$ will be added vertically and horizontally to form the initial pair representation. The initial pair representation will then be added to the projected representation of the pair feature $z_F$, with a channel size of $D_z = 64$. Thus, the output of the embedding layer contains the sequence representation $s \in R^{L \times 64}$ and the pair representation $z \in R^{L \times L \times 64}$ (Fig. 1B).

We also embedded the 1-D and 2-D positional encodings ('pos' and 'relpos' layers in Fig. 1B) to $s$ and $z$. A recycling strategy is used by encoding the geometry descriptors of the predicted structure conformation (for end-to-end models only), bringing the 1-D representation and the 2-D representation from the previous recycle to the input of the current cycle. The recycle number is set to 3 for both end-to-end and geometry models (Fig. 1A).

### RNA transformer network

There are a total of 48 transformer blocks in DRfold. The transformer block module is extended from the design of the Evoformer in AlphaFold2[18]. As shown in Fig. 1C, the sequence representation $s$ and the pair representation $z$ will first go through a sequence row-wise gated self-attention with a pair bias module that outputs the new sequence representation. The number of heads and the channel size of each head are set to 8 and 8, respectively. A sequence transition layer that contains 2 linear layers is stacked after the sequence self-attention. The sequence transition layer first expands the dimension from 64 to 128 and then projects it to the original channel size (64). The obtained sequence representation is transformed to a pair representation by an outer product mean (OPM) block. The OPM block first projects the sequence channel size to 12 with two separated linear layers. After the outer product mean operation, the output 2-D representation thus has

a channel size of 12 × 12. Another linear layer in the OPM block then projects it to the desired channel size, i.e., 64.

The 2-D representation from the OPM block will sequentially go through a set of blocks, including (1) a Triangle multiplicative update block using outgoing edges, (2) a Triangle multiplicative update block using incoming edges, (3) a Triangle self-attention block around starting node, (4) a Triangle self-attention block around starting node, and (5) a pair transition block. For the Triangle multiplicative update blocks, the channel size is set to 32; for the Triangle self-attention blocks, the number of heads and the channel size are set to 4 and 8, respectively. It should be noted that the sequence and pair blocks are stacked residually[42] for efficient and stable training.

### RNA structure module

The end-to-end model in DRfold predicts the spatial location of each nucleotide, which can be represented by a rotation matrix and a translation vector operating on a predefined conformation at a local frame. Here, we use a three-bead model, as specified by the C4', P, and glycosidic N atoms of the nucleobase (Fig. 1F), to represent the coarse-grained conformation of a nucleotide. These three atoms (C4', P, N) represent the structural centers of the nucleotide, phosphate backbone, and nitrogenous base, respectively, which are critical to determine the global structure of the RNA conformation. The full-atom models can be effectively recovered from the 3-vector virtual bond system.

The predefined conformations were obtained by collecting the resulting local structures after performing symmetric orthogonalization[43] on coordinates of each nucleotide type in the ideal A-form RNA helix structure. We assume that the three atoms form a rigid body for each of the four nucleotides. The RNA structure module takes the sequence representation from the RNA transformer network as the input to iteratively train the nucleotide-wise rotation matrices and translation vectors. As shown in Fig. 1D, the pair representation is also utilized by the invariant point attention (IPA) module to equivariantly update the RNA structure conformation during the structure module iterations. The channel sizes for the sequence and pair representations are set to 128 and 64, respectively, in the RNA structure module. The IPA hyperparameters ($N_{head}$, $c$, $N_{querypoints}$, $N_{pointvalues}$), are set to (8, 16, 4, 6), and the iteration number is set to 5.

An important step of the RNA structure module is the construction of the local frames from ground truth positions. Considering the higher flexibility of RNA structures compared to that of proteins, we construct frames with the SVD orthogonalization[43], instead of the Gram-Schmidt orthogonalization that was used in Alphafold2 (Figure S4). Since SVD orthogonalization maximizes the likelihood in the presence of Gaussian noise, it is less greedy than the Gram-Schmidt orthogonalization and thus has a lower bias during the estimation process. Levinson et al.[44] has shown that in the view of matrix reconstruction, the approximation error of SVD orthogonalization is half that of the Gram-Schmidt procedure.

### Loss function of the end-to-end models

Two types of loss functions, including the FAPE loss[18] (Fig. 1G) and the inter-N atom distance loss, are used when training the end-to-end models, i.e.,

$$L_{e2e} = 1.5 L_{FAPE} + 0.6 L_{dist} \qquad (1)$$

The FAPE loss is defined as

$$L_{FAPE} = \sum_i \sum_j \min\left(d_{cut}, \sqrt{\left\| T_i^{-1}\left(T_j(\vec{r})\right) - T_i^{exp-1}\left(T_j^{exp}(\vec{r})\right) \right\|^2 + \epsilon}\right) \qquad (2)$$

where $T_i$ (or $T_i^{exp}$) represents the Euclidean transformation, including rotation matrices and translation vectors that are learned by the networks, to convert a position $i$ at the local frame ($\vec{r}$) to the global space for the predicted model (or the target experimental structure in the training dataset). The parameter $d_{cut}$ is set to 30 Å and $\epsilon$ is set to $10^{-3}$ Å. Here, $i$ and $j$ enumerate all the nucleotide positions and all the trained atoms of the coarse-grained RNA structure, respectively. Because each term contains two reversible transforms, i.e., $T_i^{-1}\left(T_j(\vec{r})\right)$, any rigid-body transformations will be canceled out in the calculation. Therefore, $L_{FAPE}$ is by design invariant to any rigid-body conformational transformations to the predicted structures.

The distance loss function $L_{dist}$ in Eq. (1) takes the cross-entropy form of

$$L_{dist} = -\sum_{i,j} \sum_{b=1}^{38} y_{ij}^b \log p_{ij}^b \qquad (3)$$

where $y_{ij}^b$ is an indicator function to check if the distance of atom pair $(i, j)$ in the target experimental structure falls into the $b$-th distance interval; and $p_{ij}^b$ is the predicted probability for the interval. The inter-atom distance is split into 36 intervals between 2–40 Å, with two additional bins representing distances <2 Å and >40 Å.

### Prediction terms of geometry models

For a pair of residues, a set of geometry potentials are extracted from the experimentally determined structures as supervised information to train deep geometric potentials[20]. As shown in Fig. 1H, the Euclidean distance between the P, C4', and glycosidic N atoms are calculated, where the distance values for the inter-P atoms, inter-C4' atoms, and inter-N atoms are discretized into 56, 44, and 32 bins in the ranges of [2, 30 Å], [2, 24 Å], and [2, 18 Å], respectively. For each distance term, two additional bins are added representing values < 2 Å and >$M$ Å, where $M$ is the corresponding maximum distance values (30, 24, and 18 Å, respectively). Meanwhile, the long-range dihedral angles formed by atoms of each nucleotide pair $(i, j)$ are extracted, which are formed, respectively, by P($i$)-C4'($i$)-C4'($j$)- P($j$), C4'($i$)-N($i$)-N($j$)-C4'($j$), and P($i$)-N($i$)-N($j$)-P($j$). The dihedral angle values are discretized into 36 bins, plus the dimension representing whether the length of the virtual bond, i.e., C4'($i$)-C4'($j$) and N($i$)-N($j$), is larger than their corresponding maximum distance values $M$ (=24 and 18 Å, respectively).

The loss function of the geometry models is the cross-entropy loss of the distance and dihedral angle terms defined by

$$L_{geo} = -\sum_{i,j} \sum_{g \in G} w_g \log(p_{ij}^g) \qquad (4)$$

where $G$ is the set of geometry terms for the distance and dihedral angles, and the weight parameters $w_g = 1.0$ and 0.5 for the distance- and dihedral angle-related losses, respectively. The models ($p_{ij}^g$) are trained using a multi-task learning architecture as described in Fig. 1E.

### Training process of the end-to-end and geometry models

The end-to-end and geometric network models were trained using the Adam optimizer[45] with an initial learning rate of 1e−3 for 100 epochs. The maximum RNA sequence length was set to 200 following the GPU memory limits used in this study. For an RNA sequence with over 200 nucleotides, a continuous segment of 200 nucleotides was randomly sampled during the training. The batch size was set to 3 to accelerate the training with the gradient accumulation mechanism in PyTorch[46]. We also use gradient checkpointing to reduce the memory occupancy for each transformer block[47]. The whole end-to-end model was trained on a single Nvidia A40 GPU with 32GB of memory, where 6 end-to-end models and 3 geometry models with different random parameter initializations were trained, and training each of them took 2 weeks.

For the 3 geometry models, it took around 50 epochs of training for 5 days each.

## Gradient-based structure optimization with integrated end-to-end and geometry potentials

Following the end-to-end and geometry modeling, a combination of two deep-learning energy terms, $E_{DL} = E_{e2e} + E_{geo}$, is used to guide the next step of RNA structure optimization. The first end-to-end potential is written as

$$E_{e2e} = \sum_{i,j} \sum_{k \in \{P, C4', N\}} \min\left(d_{cut}, \sqrt{\left\|T_i^{-1}\left(T_j\left(\vec{r}^k\right)\right) - T_i^{conf^{-1}}\left(T_j^{conf}\left(\vec{r}^k\right)\right)\right\|^2 + \epsilon}\right) \tag{5}$$

where $T_i$ is the predicted rotation matrices and translation vectors by the end-to-end models as defined in Eq. (2), which are kept unchanged during the structure optimization. $T_i^{conf}$ represent the transforms to recover the RNA conformation of the predicted model for the atom set $\{P, C4', N\}$ from the predefined local frame, $\{\vec{r}^P, \vec{r}^{C4'}, \vec{r}^N\}$. We sum the end-to-end energy values calculated by the 6 end-to-end models as the final consensus end-to-end potential.

The geometry potential $E_{geo}$ is defined as

$$E_{geo} = -\sum_{ij}\left[\log\left(\frac{P_{ij}^d\left(d_{ij}\right) + \epsilon}{P_{ij}^{dN} + \epsilon}\right) + 0.5 * \log\left(P_{ij}^\theta\left(\theta_{ij}\right) + \epsilon\right)\right] \tag{6}$$

where $P_{ij}^d\left(d_{ij}\right)$ and $P_{ij}^\theta\left(\theta_{ij}\right)$ are the predicted probabilities for a given distance $d_{ij}$ and dihedral angles $\theta_{ij}$ between a nucleotide pair $(i,j)$; $P_{ij}^{dN}$ and $P_{ij}^{\theta N}$ are the corresponding probabilities of the last distance bin below the upper threshold. The negative log-likelihood of the predicted probabilities is interpolated using cubic spline interpolation to form a potential curve for the specific distance/dihedral angle terms[20].

The 6 conformations predicted by the end-to-end models are also used as initial structures for the optimization system and separately optimized by the same hybrid potential function. The gradient of parameters with respect to the hybrid potential function can be calculated by the automatic differentiation package in PyTorch. Given the energy values and gradients, we can use the L-BFGS algorithm[48] to iteratively update the parameters of the system, i.e., $T_i^{conf}$, which determines the 3D conformations of the RNA models. The conformation with the lowest energy is considered as the final predicted structure among the 6 different L-BFGS trajectories.

## Atomic-level structure refinement

Since both the end-to-end pipeline and the L-BFGS folding simulations operate on the reduced 3-bead model, we implement a two-step procedure to reconstruct and refine the full atomic model of DRfold. During the first step, we use Arena (https://zhanggroup.org/Arena/) to construct the standard conformations of the full-atomic structure for each of the four types of nucleotides, based on the generic A-form RNA helix with a 32.7° twist and a 2.548 Å rise[49], which are then superimposed to the three-atom frame (P, C4', N) of the coarse-grained DRfold model to quickly obtain the initial full-atomic RNA structure. In this step, three fast refinement steps are taken to rectify incorrect bond lengths and angles, base and base pair conformations, and atom clashes, respectively, while keeping the input 3-bead model frozen. Next, a full-atom MD minimization is performed using OpenMM[35] to further refine the local structure geometry, including steric clash and bond-length/angle violation removal. The MD force field is based on AMBER14[36] and specified by 'amber14-all.xml' and 'amber14/tip3pfb.xml' in the package. For each model, $k \times N_{atoms}/20$ minimization steps are run, where $k = 0.6$ is the empirical coefficient and $N_{atoms}$ is the number of atoms in the full-atomic structure.

## Reporting summary

Further information on research design is available in the Nature Portfolio Reporting Summary linked to this article.

## Data availability

The data supporting the findings of this study are available from the corresponding authors upon reasonable request. All input data are freely available from public sources. We show structures of 7OX9 and 7MRL obtained by four-digit accession codes in the Protein Data Bank repository (https://www.rcsb.org/). RNA structures for training were collected from PDB (https://www.wwpdb.org/ftp/pdb-ftp-sites).

## Code availability

The DRfold standalone package are available at https://zhanggroup.org/DRfold/ and https://github.com/leeyang/DRfold/. Data were analyzed using Numpy v.1.20.3 (https://github.com/numpy/numpy), SciPy v.1.7.1 (https://www.scipy.org/), and Matplotlib v.3.4.3 (https://github.com/matplotlib/matplotlib). Structures were visualized by Pymol v.2.3.0 (https://github.com/schrodinger/pymol-open-source) and UCSF ChimeraX v.1.5 (https://www.cgl.ucsf.edu/chimerax/).

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

## Acknowledgements

We thank Drs. Sha Gong and Xi Zhang for insightful discussions. This work was supported in part by the National Institute of General Medical Sciences (GM083107, GM116960, GM136422 to Y.Z.); the National Institute of Allergy and Infectious Diseases (AI134678 to P.L.F. and Y.Z.); the National Institute of Health Office of The Director (OD026825 to Y.Z.); the National Science Foundation (DBI2030790 and IIS1901191 to Y.Z., MTM2025426 to P.L.F. and Y.Z.). Start-up Grants of the National University of Singapore (A-8001129-00-00, A-0001166-36-00, A-8000974-00-00 to Y.Z.). This work used the Extreme Science and Engineering Discovery Environment (XSEDE), which is supported by National Science Foundation (ACI1548562).

## Author contributions

Y.Z. conceived and designed the project and supervised the work. Y.L. developed the methods and performed the benchmark. C.Z. prepared the data, developed the full atom packing software, and performed the benchmark. C.F. prepared the initial data and performed the benchmark. R.P. prepared data and participated in discussions. P.L.F. supervised the work. All authors wrote the manuscript and approved the final version.

## Competing interests

The authors declare no competing interests.
