## [Peer Review File · Nature Communications]

Integrating end-to-end learning with deep geometrical potentials for ab initio RNA structure predictionREVIEWER COMMENTS

Reviewer #1 (Remarks to the Author):

The authors present a new method for predicting the 3D structure of RNA. It uses a deep learning model and predicts the tertiary structure based on sequence and secondary structure in the end-to-end approach. The authors present the scheme and performance of the method and compare the models it generates to predictions obtained from 3 other computational methods. The new method was tested on a collection of 40 small RNA molecules.

The paper is written quite clearly. However, I have a few comments and observations. The main drawback of the paper is the very modest experimental part. The authors have chosen 3 existing methods to which they compare their algorithm. Among them, there is no new method based on deep learning (at least 4 are available). The selection of control methods is also questionable. In addition, a pressing problem in RNA bioinformatics is currently the prediction of large structures, pseudoknots, and non-canonical pairings. DRfold does not touch these problems - why? One final key point: predictions are not validated for the correctness of structural parameters.

Detailed questions and comments are given below:

Major:

1) "Despite the advantage that they do not directly rely on 58 templates, the accuracy of the fragment assembly and ab initio folding approaches is low in general." - this is an interesting statement but seems not to have been supported by computational experiments. The results of the RNA-Puzzles and CASP-RNA competitions do not show that these approaches are particularly inaccurate. It is possible that the authors had in mind all 3D prediction methods and not just those based on fragment assembly and ab initio modeling...

2) I do not understand what the first embedding layer does. Its input is sequence and secondary structure, and the output is the same (sequence and base pairs). Does it simply transform one representation of data into the other? What representations are accepted at the input, and which ones are given as output?

3) The test set includes only small RNAs (14-392nts). Their computational prediction is not a big problem. Why did the authors limit the experiment to small RNAs? All RNA specialists agree that the prediction of large RNA structures is a challenge.

4) How did the authors ensure that there was no redundancy in the dataset? At what structural level was the redundancy determined: sequence, 2D structure, or 3D structure? In creating this dataset, were the authors assisted by data from the BGSU RNA Hub (<https://www.bgsu.edu/research/rna/databases/non-redundant-list.html>)? Did they take into account representatives of Rfam families (<https://rfam.org/>)? Did they use ready-made non-redundant

collections from the RNAsolo database (<https://rnasolo.cs.put.poznan.pl/archive/>)?

5) RNA-BRiQ is quite a new method, not widely tested in RNA community initiatives like RNA-Puzzles or CASP-RNA, so its quality and accuracy have not been proven in blind experiments. I suggest not using such tools as control methods. As a reference method applying the ab initio approach, you should use SimRNA (Boniecki et al., 2016; doi:10.1093/nar/gkv1479) or FARFAR (Watkins et al., 2020; doi:10.1016/j.str.2020.05.011). SimRNA also follows coarse-grained computation, so it seems an obvious choice to compare.

6) "Among the 40 test targets, 7 targets were found to have been 117 successfully folded by DRfold at atomic resolution (RMSD < 2.5 Å)" - RMSD here means full-atom RMSD or RMSD calculated based on P atoms only? I understand from the description that all RMSDs are computed for P atoms. In that case, the term "atomic resolution" is an abuse. Very misleading. Otherwise, if the RMSD is given for all atoms, then the authors should specify which RMSD values were computed for all atoms and which for P atoms solely.

7) RMSD, although commonly used, is not a good measure for assessing the quality and accuracy of structure prediction. There are better measures dedicated to RNA molecules. These are INF - Interaction Network Fidelity and DI - Deformation Index (Parisien et al., 2009; doi:10.1261/rna.1700409). These measures are commonly used in the RNA community, in RNA-Puzzles and CASP-RNA competitions. RNA-Puzzles also uses MCQ - a measure based on torsion angles (Zok et al., 2014; doi:10.1007/s10100-013-0296-5). I request that the authors perform the comparative analysis again and determine the values of these measures (INF - split into canonical and non-canonical base pairs, DI, MCQ). Tools and scripts to compute all these measures are available in the RNA-Puzzles toolkit (Magnus et al., 2020; doi:10.1093/nar/gkz1108). I am particularly interested in the accuracy of predicting non-canonical interactions.

8) There are some DL-based models to predict RNA 3D structure. I wonder why DRfold has not been compared with these methods... In my opinion, the section on comparative analysis is minimalist. Most of the existing prediction methods have not been taken into account in comparison. No comparison was made with recent approaches based on DL models. Other DL methods for RNA 3D structure prediction:

- FARFAR+ARES (<https://pubmed.ncbi.nlm.nih.gov/34446608/>)
- DeepFoldRNA (<https://doi.org/10.1101/2022.05.15.491755>)
- E2Fold-3D (<https://doi.org/10.48550/arXiv.2207.01586>)
- <https://doi.org/10.1101/2021.08.30.458226>

The methods are available on GitHub.

9) In addition to the evaluation based on comparing predictions with the reference structures, was any validation of the predicted models done? I mean, for example, checking clash-score, torsion angles, bond lengths, the handedness of helices, etc... Analysis of computer-generated models (Carrascoza et al., 2022; doi:10.1261/rna.078685.121) shows that prediction methods rarely implement functions to check and correct stereochemical parameters. Do the models built by DRfold have the correct structure

parameters?

10) There are several coarse-grained models developed for RNA. Why did the authors choose this particular 3-bead model?

Minor:

- "For examples, methods" -> "For example, methods"
- "1D sequence" is an awkward phrase, sequence is 1D by default, 1D structure is an alternative term for the sequence, so use "sequence" or "1D structure"
- "structurewhere, where" -> "structure, where"
- "RNAComposor" -> "RNAComposer"
- "other top methods which requires" -> "other top methods which require"

Reviewer #2 (Remarks to the Author):

Inspired by the success of deep learning method in protein 3D structure prediction of protein, its application to that of RNA is expected. DRfold is such a method. It uses similar deep learning framework as AlphaFold2 but has some novel features such as using query sequence and predicted secondary structures as input, using coarse-grained end-to-end learning, using two independent transformer blocks to learn local features and geometric restraints of RNA 3D structure simultaneously and converts them into a composite potential to optimize the three-atom coarse-grained structure of RNA. DRfold outperforms the control methods on the test set and also showed good performance in CASP15. DRfold provides a new approach to predict RNA 3D structure and I recommend it for publication after some minor revisions.

1. For the control methods, it should clearly give what their inputs are. For examples, what secondary structures are used by RNAComposer and 3dRNA in the prediction? Predicted by RNAfold or PETfold? What procedure is used in the prediction by 3dRNA? Assemble or Optimize?
2. In DRfold, the secondary structure is predicted by two complementary methods: RNAfold and PETfold. How is the predicted SS matrix obtained from the results both of them? Since PETfold uses MSA as input, does the result of DRfold also depend on MSA?
3. Modified nucleotides can be mutated into standard nucleotides, why are they classified separately? or what does DRfold do with RNAs containing "N"? Is it mutated to a standard base or removed?
4. In line 371, how is the predefined conformation specified for each nucleotide?
5. The RNAs in the training set are collected from sequence cluster centers (sequence identity cutoff of 90%) of solved structures deposited in or after the year 2021 in the PDB : is the prediction accuracy sensitive to the cutoff? For example, 80%.

Reviewer #3 (Remarks to the Author):

This paper demonstrates that ab initio RNA 3D structure prediction can be achieved using end-to-end deep learning similar to AlphaFold2, and that the prediction accuracy is better than that of existing RNA 3D structure prediction methods. Considering that the number of RNA 3D structures available as training data is an order of magnitude smaller than that of proteins, it is very reasonable to adopt a coarse-grained three-atom model instead of a full-atom model like AlphaFold2, and to incorporate the RNA secondary structure predictions. It is unfortunate that even with these innovations, the results of the blind test on CASP15 still fall short of the existing approaches.

p.3, l.60: "SPOT-RNA and e2efold", p.7, l.245: "Previous studies 8,9":

In this paper, these two methods are listed as methods for RNA secondary structure prediction using deep learning. However, e2efold has been shown in later studies to suffer from severe overfitting and is inappropriate to be listed as a representative study; MXfold2[1] and Ufold[2] would be more appropriate.

[1] Sato et al. 2021. "RNA Secondary Structure Prediction Using Deep Learning with Thermodynamic Integration." *Nature Communications* 12 (1): 941.

[2] Fu et al. 2022. "Ufold: Fast and Accurate RNA Secondary Structure Prediction with Deep Learning." *Nucleic Acids Research* 50 (3): e14.

p.4, l.96: "sequence identify cutoff of 90%"

Sequences with 90% sequence identity are extremely similar, and with such similar sequences in the training data, it is not surprising how easy it is to predict the structure. Since many benchmarks for RNA secondary structure prediction use a cutoff of 80% sequence identity, it is necessary to set the cutoff at least at the same level. You should also indicate how homologous the 40 sequences in the test data are to the training data, and whether there is a correlation between homology to the training data and the accuracy of the 3D structure prediction.

p.4, l.100: structurewhere → structure

p.4, l.105: "DRfold outperforms current RNA structure predictors."

This section shows that the proposed method has good prediction accuracy compared to existing RNA 3D structure methods without deep learning. However, in recent years, several deep learning-based RNA 3D structure prediction methods similar to the proposed method have been reported. It is necessary to compare the proposed method with them.

- Feng et al. 2022. "Accurate de Novo Prediction of RNA 3D Structure with Transformer Network." *bioRxiv*. <https://doi.org/10.1101/2022.10.24.513506>.

- Zhang et al. 2022. "Physics-Aware Graph Neural Network for Accurate RNA 3D Structure Prediction." *arXiv [cs.LG]*. arXiv. <http://arxiv.org/abs/2210.16392>.

- Baek et al. 2022. "Accurate Prediction of Nucleic Acid and Protein-Nucleic Acid Complexes Using RoseTTAFoldNA." *bioRxiv*. <https://doi.org/10.1101/2022.09.09.507333>.

- Pearce et al 2022. "De Novo RNA Tertiary Structure Prediction at Atomic Resolution Using Geometric

Potentials from Deep Learning.” *bioRxiv*

. <https://doi.org/10.1101/2022.05.15.491755>.

- Shen et al. 2022. “E2Efold-3D: End-to-End Deep Learning Method for Accurate de Novo RNA 3D Structure Prediction.” *arXiv [q-bio.QM]*. arXiv. <http://arxiv.org/abs/2207.01586>.

p.5, l.150: “Interestingly, after excluding the 5 extreme targets with length > 200, the PCC becomes extraordinarily different, i.e., 0.71 (see dashed line in Figure 2D). Such correlation would suggest that the performance will be better with longer (< 200) RNA targets.”

The linear regressions in Figure 2D do not even appear to be a good fit because the prediction accuracies are so scattered. A statistical test on the regression coefficient will tell us whether these regressions are a good fit or not. In any case, the above statement is overly optimistic because it is an extrapolation for the portion of the sequence length that exceeds 200.

p.7, l.242: “the significant importance of the SS embedding feature”

Given the importance of RNA secondary structure, it would be nice to have a comparison not only with the combination of RNAfold and PETfold, but also with other prediction methods or with the correct secondary structures obtained from the correct 3D structures given.

p.7, l.252: “Blind RNA structure prediction in CASP15”

This section reports the results of blind test on CASP15. The discussion here is based on the z-score of RMSD and TM score for relative accuracy comparison with other methods. However, to determine the generalization ability of the proposed method, a comparison with experiments on the training and test data prepared in this paper based on the raw RMSD and TM score is necessary.

Response to Reviewer #1

We very much appreciate the comments and suggestions from the Reviewer, which we found very helpful for improving the quality of the DRfold program and manuscript. The major concern from the Reviewer involves the lack of comparison with the recently developed deep-learning approaches. In the revision, we have added five state-of-the-art deep learning methods, including ARES, DeepFoldRNA, RhoFold, RoseTTAFoldRNA, and trRosettaRNA, for more comprehensive benchmark tests. The Reviewer also raised concerns on the examination of local structural validation of the DRfold models. Accordingly, we have added multiple experimental analyses to quantitatively examine the local structural quality, where we found that a newly added structural refinement stage is important for improving the atomic-level quality and physical realism of the DRfold models. In addition, we have carefully addressed other important concerns raised by the Reviewer, including target size of the test RNAs, redundancy settings, and method parameter clarifications. In the following, we include point-by-point replies to the comments of the Reviewer, where all changes have been highlighted in yellow in the manuscript.

1. The Reviewer commented:

The authors present a new method for predicting the 3D structure of RNA. It uses a deep learning model and predicts the tertiary structure based on sequence and secondary structure in the end-to-end approach. The authors present the scheme and performance of the method and compare the models it generates to predictions obtained from 3 other computational methods. The new method was tested on a collection of 40 small RNA molecules.

We appreciate the summary from the Reviewer on the work.

2. The Reviewer commented:

The paper is written quite clearly. However, I have a few comments and observations. The main drawback of the paper is the very modest experimental part. The authors have chosen 3 existing methods to which they compare their algorithm. Among them, there is no new method based on deep learning (at least 4 are available). The selection of control methods is also questionable.

Thank you for raising this important point. In the revised manuscript, we have compared the performance of DRfold with 4 publicly available third-party deep learning-based methods, i.e., ARES, RhoFold, RoseTTAFold2NA and trRosettaRNA, as well as a program DeepFoldRNA previously developed in our lab (see **Response 12** below for more details). In addition, two traditional methods, SimRNA and FARFAR2, are also considered as control methods in the revised manuscript (see **Response 9** below for more details).

3. The Reviewer commented:

In addition, a pressing problem in RNA bioinformatics is currently the prediction of large structures, pseudoknots, and non-canonical pairings. DRfold does not touch these problems - why?

Thank you for pointing out the pressing problems in RNA structure prediction. First, we agree that the structure prediction for large RNAs is a significantly challenging problem in RNA structure prediction. Accordingly, we added analyses and comments on the limited performance of DRfold on modeling large-size RNA structures in both the Results and Discussion sections. Specifically, we found that one of the major limits on large RNAs comes from the fact that the current models were trained on small RNAs below 200 NTs (due to the GPU memory limit) (see **Response 7** below for a detailed discussion).

Second, we added multiple experiments to carefully examine the ability of DRfold in modeling pseudoknots, non-canonical base-pairings, and other secondary structure aspects in Page 6-7 with results summarized in Tables S1-2 and Figure S2. A detailed discussion of these issues is presented in **Response 11** below.

4. The Reviewer commented:

One final key point: predictions are not validated for the correctness of structural parameters.

We thank the Reviewer for raising this important point. In the revised manuscript, we introduced a new two-step procedure for atomic structure refinement. Accordingly, we added a new section entitled “**Structure refinement improves physical realism of DRfold models**” to discuss the validation results of the correctness of the structural parameters. We found that the newly introduced refinement procedure is critical to improve the validation and physical realism of the DRfold models (See **Response 13** below for detailed explanation).

5. The Reviewer commented:

Detailed questions and comments are given below:

Major:

1) "Despite the advantage that they do not directly rely on 58 templates, the accuracy of the fragment assembly and ab initio folding approaches is low in general." - this is an interesting statement but seems not to have been supported by computational experiments. The results of the RNA-Puzzles and CASP-RNA competitions do not show that these approaches are particularly inaccurate. It is possible that the authors had in mind all 3D prediction methods and not just those based on fragment assembly and ab initio modeling...

We thank the Reviewer for pointing out the ambiguous statement in the manuscript. We made the statement mainly on our benchmark results using default settings and automated implementations. However, we agree that in RNA-Puzzles and CASP experiments, many of these methods had reasonable performance, which we presume is partly due to the fact that the groups utilized human intervention in model tweak and selection during the competitions which may help improve their apparent performance. To clarify, we rewrote the sentences as (Page 3):

Another family of methods, typified by RNAComposer⁴ and 3dRNA⁵, assemble full-length RNA structures from fragments searched from a prebuilt fragment library. *Ab initio* RNA structure prediction methods, such as SimRNA⁶, FARFAR2⁷ and RNA-BRiQ⁸ apply statistical potentials to guide the structure folding simulations. **Although utilizing domain expert knowledge with these methods could lead to somewhat better performance in RNA-Puzzles and CASP^{9, 10}, their**

performance is often suboptimal in automated benchmark test runs, suggesting that automatic prediction of regular RNA structures remains a challenging task to the *ab initio* simulations.

6. The Reviewer commented:

2) I do not understand what the first embedding layer does. Its input is sequence and secondary structure, and the output is the same (sequence and base pairs). Does it simply transform one representation of data into the other? What representations are accepted at the input, and which ones are given as output?

Thank you for raising the question. The embedding layer uses a set of linear layers to transform the input features, i.e., sequence feature $s_F \in R^{L \times 5}$ and the pair feature $z_F \in R^{L \times L \times 4}$ into the sequence representation $s \in R^{L \times 64}$ and the pair representation $z \in R^{L \times L \times 64}$, both of which are not identical to the input. The hidden representations ($s \in R^{L \times 64}$ and $z \in R^{L \times L \times 64}$) are needed to serve as the input of the subsequent RNA transformer networks. Here, 64 is a hyper-parameter. We have added the following details into the second paragraph of **Feature preparation and embedding** to clarify the point (Page 13):

The **query** sequence feature $s_F \in R^{L \times 5}$ and the pair feature $z_F \in R^{L \times L \times 4}$ act as the input of the embedding layer. Here, the embedding layer is a neural network module that transforms the input features (s_F and z_F) into learned representations (sequence representation s and pair representation z) through a set of linear layers. After the embedding layer, each residue and residue-residue pair will be represented by a 64-D vector. This layer of embedding is important for generating hidden representations (s and z) as the input for the subsequent RNA transformer networks. Specifically, the sequence feature s_F will be projected to the desired dimension ($D_s=64$) by a linear layer. Another two linear projections of s_F will be added vertically and horizontally to form the initial pair representation. The initial pair representation will then be added to the projected representation of the pair feature z_F , with a channel size of $D_z=64$. Thus, the output of the embedding layer contains the sequence representation $s \in R^{L \times 64}$ and the pair representation $z \in R^{L \times L \times 64}$ (**Figure 1B**).

7. The Reviewer commented:

3) The test set includes only small RNAs (14-392nts). Their computational prediction is not a big problem. Why did the authors limit the experiment to small RNAs? All RNA specialists agree that the prediction of large RNA structures is a challenge.

We agree that the prediction of large RNA structures is a challenge. The reason we chose this relatively small RNA range is that it is largely consistent to the RNA size range that DRfold has been trained for (i.e., <200 NTs) due to the GPU memory limitation. We have added the following sentences to clarify the point (Page 4):

The length range of the test RNAs [14, 392 Nts] was slightly larger than but generally consistent with the crop-size range of the RNAs that were used for training the DRfold models (i.e., <200 NTs) due to the GPU memory limitations.

It is noted that DRfold was trained on limited GPU resources (i.e., a single Nvidia A40 GPU with 32 GB memory) for the proof of concept. We believe that a more comprehensive training with

larger computing resources will help further improve the modeling results especially for large-sized RNAs. We have discussed this issue in the following paragraph (Page 6):

It is notable that for those targets with lengths > 200 NTs, the TM-scores obtained by DRfold are lower overall than those obtained for smaller targets < 200 NTs. One reason for the suboptimal performance for large-size RNAs is probably that a maximum RNA length cutoff was set to 200 NTs when we trained the models in DRfold due to the limited GPU memory (with a single Nvidia A40 GPU with 32 GB memory) used during the training, and therefore the interaction patterns for extremely distant (>200) nucleotide pairs may not be sufficiently learned. Developing length-insensitive variants of attention networks by utilizing more comprehensive RNA dataset and larger computing resources should help DRfold to learn the longer-range inter-residue interactions and therefore enhance its ability to fold large-sized RNA structures.

Finally, we also emphasized the challenge of modeling large RNAs in the Discussion (Page 12):

The second limitation of DRfold comes from the present GPU memory resource which allows model training only on small RNAs <200 NTs and, together with the limitation of the training structural repositories, made the current models fall short on large-size RNA modeling as demonstrated in both the benchmark and CASP15 experiments.

8. The Reviewer commented:

4) How did the authors ensure that there was no redundancy in the dataset? At what structural level was the redundancy determined: sequence, 2D structure, or 3D structure? In creating this dataset, were the authors assisted by data from the BGSU RNA Hub (<https://www.bgsu.edu/research/rna/databases/non-redundant-list.html>)? Did they take into account representatives of Rfam families (<https://rfam.org/>)? Did they use ready-made non-redundant collections from the RNAsolo database (<https://rnasolo.cs.put.poznan.pl/archive/>)?

Thank you for raising these questions. We have ensured non-redundancy of our data mainly by requesting a uniform sequence identity cutoff of 90%. We did not use BGSU and RNAsolo as they both use a sequence identity cutoff of 95%, which is slightly higher than what we are using.

Here, the non-redundancy requirement includes a sequence identity filter for both RNAs within the test dataset and the RNAs between the test and the training datasets. As mentioned in the following paragraph, we utilized the sequence identity cutoff 90% for both cases (Page 4):

DRfold was tested on 40 non-redundant RNA structures with lengths from 14 to 392 nucleotides, which were collected from sequence cluster centers (sequence identity cutoff of 90%) of solved structures deposited in or after the year 2021 in the PDB²¹. Structures without any valid base pairs were not included. All test sequences, as well as their cluster members, were excluded from the training dataset, which contains 3864 unique RNAs extracted from the PDB that were deposited before the year 2021. Thus, a filter based on both sequence identity and time stamp implements a stringent separation between the training and testing datasets of DRfold, both of which are available for download at <https://zhanggroup.org/DRfold>.

Since the second filter between test and training datasets is particularly important for reducing the danger of overfitting of the machine-learning models, we added three additional filters at sequence identity cutoffs of 80%, 70% and 60% respectively. It was observed that although, like most of other deep learning methods, the performance of DRfold decreases with more stringent sequence

identity cutoffs, the DRfold models consistently outperform the control methods. We have added the following paragraph to summarize the results (Page 5):

To investigate the possible impact of sequence homology cutoffs on the accuracy of the DRfold models, an exhaustive test considering various sequence identity cut-offs between the training and test sets was conducted. Following conventional criteria used in previous studies for RNA structure/SS prediction using deep learning^{11, 23, 24, 25}, additional datasets were constructed by excluding targets with sequence identities greater than multiple thresholds (i.e., 80%, 70% and 60%) to the DRfold training dataset; this resulted in 32, 23, and 10 sequence-nonredundant RNA structures, respectively. The results show that the performance of DRfold correlates with the sequence cut-offs for the test sets, where the average TM-scores of the selected test targets gradually decreased from 0.435 to 0.309 as the maximum sequence identity cut-off decreased from 90% to 60% (**Figure 2D**). These data suggest that the deep learning-based predictions are more reliable when trained on similar sequences. Nevertheless, the average TM-score for DRfold consistently exceeded that of the best control methods by at least 33.9% across all thresholds.

Figure 2D. The correlation between the TM-score and the sequence identity cut-off.

9. The Reviewer commented:

5) RNA-BRiQ is quite a new method, not widely tested in RNA community initiatives like RNA-Puzzles or CASP-RNA, so its quality and accuracy have not been proven in blind experiments. I suggest not using such tools as control methods. As a reference method applying the ab initio approach, you should use SimRNA (Boniecki et al., 2016; doi:10.1093/nar/gkv1479) or FARFAR (Watkins et al., 2020; doi:10.1016/j.str.2020.05.011). SimRNA also follows coarse-grained computation, so it seems an obvious choice to compare..

Thank you for the constructive suggestion. We have added now SimRNA and FARFAR to our control method set. Since BRiQ was used by one of the top performing teams in the CASP15 RNA structure prediction, [AIchemy \(https://www.predictioncenter.org/casp15/doc/CASP15_Abstracts.pdf#AIchemy-RNA2\)](https://www.predictioncenter.org/casp15/doc/CASP15_Abstracts.pdf#AIchemy-RNA2), we still kept the results of BRiQ in the tables, which should help provide additional reference together with the newly included methods. Accordingly, we have updated the data in Figures 2 and 3 and

rewritten the following paragraphs of **DRfold outperforms previous RNA structure predictors** (Page 4-5):

To benchmark the performance of DRfold with previous approaches, two representative fragment assembly methods, RNAComposer⁴ and 3dRNA⁵, and three representative *ab initio* RNA structure prediction methods, RNA-BRiQ⁸, SimRNA⁶ and FARFAR2⁷, were considered as control methods, where a brief introduction to the configurations of these methods is given in **Text S1**. **Figure 2A** compares the root mean squared deviation (RMSD) of the models generated by DRfold and the control methods relative to the target structures, where the coordinates of the P atoms were used for topology evaluation. The average RMSD value obtained by our method (14.45 Å) was significantly lower than those obtained by 3dRNA (20.54 Å), FARFAR2 (22.48 Å), RNAComposer (20.80 Å), BRiQ (22.88 Å) and SimRNA (23.88 Å), where the corresponding *P*-values obtained by Student's *t*-tests were 7.35E-05, 3.72E-07, 1.90E-04, 3.34E-07 and 6.14E-07, respectively. The median RMSD of DRfold was 9.38 Å, compared to the lowest median RMSD of 19.04 Å obtained by the control methods (RNAComposer). Among the 40 test targets, 6 (or 2) targets were found to be successfully folded by DRfold at a high accuracy with RMSDs < 2.5 Å, as evaluated by the P-atom (or full-atom) RMSD. In **Figure 2B**, we further list the accumulative fraction of cases with RMSD values below thresholds ranging from 2.5 Å to 15.0 Å, where DRfold generated significantly more cases than the control methods across all RMSD cutoffs. For example, 47.5% of the DRfold models had an RMSD less than 7.5 Å, which is more than twice the fraction (20.0%) obtained by the best-performing third-party method, 3dRNA.

Since a local error could cause a high RMSD, the RMSD value may not be ideal for assessing the quality of the RNA models at the high RMSD range. In **Figures 2C**, we further list the results for the TM-score, an index that is more sensitive to the global fold of the RNA structures²². Here, TM-score ranges from (0,1] with a higher value indicating a closer structural similarity, where a TM-score above 0.45 indicates a correct fold for RNA structures independent of the sequence length. As shown in **Figure 2C**, the average TM-score of the DRfold models (0.435) was 73.3% higher than the average TM-score of 0.251 obtained by the second-best method, 3dRNA, with a *P*-value of 5.79E-07. Furthermore, 45% (=18/40) of the DRfold models had correct folds with TM-scores > 0.45, while the second-best method only achieved a success rate of 12.5%. The ability of DRfold to obtain very high-quality overall models for a substantial fraction of targets is apparent in the large upper shoulder in the distribution of TM-scores shown in **Figure 2C**.

In **Figure 3**, we present a detailed head-to-head comparison of TM-score and RMSD between DRfold and the control methods, where a pronounced advantage of DRfold with the control methods was observed in all boxes. For example, the fraction of the test targets for which DRfold achieved a lower RMSD than the control methods was 80.0% (to 3dRNA), 82.5% (FARFAR2), 72.5% (RNAComposer), 75.0% (RNA-BRiQ) and 80.0% (SimRNA), respectively, as shown in **Figures 3A-E**. The superiority of DRfold was more robust when evaluated by TM-score in **Figures 3F-J**, as the maximum absolute difference in TM-score was only 0.039 for those targets where any of the control methods performed better. In contrast, for those targets where DRfold had a higher TM-score, the absolute difference was up to 0.734. Such observation suggests again that DRfold can consistently predict better global conformations compared to the classic RNA folding methods.

10. The Reviewer commented:

6) "Among the 40 test targets, 7 targets were found to have been 117 successfully folded by DRfold at atomic resolution (RMSD < 2.5 Å)" - RMSD here means full-atom RMSD or RMSD calculated based on P atoms only? I understand from the description that all RMSDs are computed for P atoms. In that case, the term "atomic resolution" is an abuse. Very misleading. Otherwise, if the RMSD is given for all atoms, then the authors should specify which RMSD values were computed for all atoms and which for P atoms solely.

Thank you for the question, which help us clarify the issue. Here the RMSD values were calculated by P atoms only. In the revised manuscript, an atomic-level refinement procedure has been added to DRfold. Consequently, after the refinement there are now 6 targets with P atom RMSD < 2.5 Å. We also computed the full-atom RMSD and found that two targets were folded at atomic resolution (full-atom RMSD < 2.5 Å). Accordingly, we have updated the paragraph as follows (Page 4):

Among the 40 test targets, **6** (or **2**) targets were found to be successfully folded by DRfold **at a high accuracy with RMSDs < 2.5 Å, as evaluated by the P-atom (or full-atom) RMSD.**

11. The Reviewer commented:

7) RMSD, although commonly used, is not a good measure for assessing the quality and accuracy of structure prediction. There are better measures dedicated to RNA molecules. These are INF – Interaction Network Fidelity and DI - Deformation Index (Parisien et al., 2009; doi:10.1261/rna.1700409). These measures are commonly used in the RNA community, in RNA-Puzzles and CASP-RNA competitions. RNA-Puzzles also uses MCQ - a measure based on torsion angles (Zok et al., 2014; doi:10.1007/s10100-013-0296-5). I request that the authors perform the comparative analysis again and determine the values of these measures (INF - split into canonical and non-canonical base pairs, DI, MCQ). Tools and scripts to compute all these measures are available in the RNA-Puzzles toolkit (Magnus et al., 2020; doi:10.1093/nar/gkz1108). I am particularly interested in the accuracy of predicting non-canonical interactions.

We thank the Reviewer for pointing out these model assessment criteria. Following the suggestion, we have added the following paragraph to discuss the performance of DRfold and the control methods in terms of interaction network fidelity (INF) and deformation index (DI). Here, INF evaluation was split into the categories INF_wc, INF_nwc and INF_stack. The results show that DRfold consistently outperforms the controls methods for all different assessment criteria (Page 6):

The hydrogen-bonding interactions between conjugated nucleotides are critical to stabilize the tertiary structures and functions of RNAs. It is therefore useful to investigate whether and how DRfold can recover these SS patterns. In **Table S1**, we summarize the base interaction network fidelity (INF)^{26, 27} and deformation index (DI)²⁸ scores of the models generated by DRfold and the control methods, which were calculated using the RNA-Puzzles toolkit²⁹. Here, the INF was split into four categories, including Watson-Crick (INF_wc), non-Watson-Crick (INF_nwc), stacking (INF_stack), and overall interactions (INF_all). Although DRfold does not employ specific base-pairing related potentials, it outperforms other methods across each evaluation index, suggesting that the relative frame positions in the frame aligned point error (FAPE) and geometrical potentials may have implicitly helped DRfold to recover the base pairing patterns of the structure models (see **Methods**). In the lower panel of **Table S1**, we also list the performance comparisons on the targets with a sequence identity cut-off of 80% to the DRfold training set, where DRfold still showed an advantage compared to the automatic control methods. It is noted that despite the overall advantage, the success rate of non-canonical base pairing prediction (INF_nwc) by DRfold was still low. A more detailed learning model at the atomic level trained on the datasets with enhanced non-canonical pairing samples might help improve the accuracy for INF_nwc.

Table S1. Comparison of RNA base pairing related metrics between DRfold and control methods in the test set with a sequence identity cutoff of 90% and 80% to the training set respectively. The bold fonts highlight the best performing value in each category.

Methods	DI	INF_all	INF_wc	INF_nwc	INF_stack
---------	----	---------	--------	---------	-----------

Sequence identity cutoff <90%					
3dRNA	40.30	0.586	0.633	0.067	0.597
FARFAR2	43.92	0.604	0.589	0.042	0.644
RNAComposer	41.17	0.616	0.638	0.142	0.628
RNA-BRiQ	39.36	0.622	0.613	0.095	0.645
SimRNA	50.52	0.528	0.384	0.012	0.616
DRfold	26.27	0.708	0.768	0.155	0.711
Sequence identity cutoff <80%					
3dRNA	36.68	0.581	0.625	0.046	0.593
FARFAR2	41.72	0.596	0.575	0.033	0.638
RNAComposer	46.76	0.605	0.616	0.122	0.622
RNA-BRiQ	43.41	0.616	0.599	0.069	0.641
SimRNA	40.90	0.525	0.377	0.000	0.612
DRfold	31.63	0.673	0.722	0.126	0.678

In Table S2, we also evaluated the DRfold models in terms of MCQ and a newly defined Handedness score to examine the torsion angle and chirality of the RNA models. The following paragraph has been added to summarize the results (Page 6):

In Table S2, we further list the mean of circular quantity (MCQ)³¹ and Handedness scores of the DRfold models compared to the five control methods. Here, the MCQ score measures the dissimilarity between two structures in torsion angle space using full-atom representations³¹, while the Handedness score evaluates the correctness of the chirality of the RNA helices and computes the fraction rate of non-loop residues in the predicted models that have closer C4' torsion angles to the targets than to the mirror images of the target structures. The results show that while DRfold does not outperform the control methods in MCQ, likely due to the coarse-grained representation of RNA structures during the model training procedure, DRfold excels in Handedness score over all control methods. This performance could be attributed to the use of mirrored-sensitive potentials by DRfold, including the reflection transformation-sensitive FAPE potential¹⁸ and the use of long-range dihedral angles in the geometry potential, which are capable of distinguishing the desired structures from their mirror images.

Table S2. Comparison of RNA local torsion angle parameters between DRfold and the control methods. The bold fonts highlight the best performing value in each category.

Methods	MCQ	Handedness score
3dRNA	0.598	0.636
FARFAR2	0.491	0.616
RNAComposer	0.533	0.611
RNA-BRiQ	0.449	0.648
SimRNA	0.531	0.563
DRfold	0.553	0.730

Finally, we added the following paragraph and Figure S2 to summarize the ability of DRfold on modeling pseudoknot structures (Page 7):

Out of the 40 test targets, 15 contain pseudoknots as assigned by DSSR³⁰. While 3dRNA and RNAComposer cannot detect any of the pseudoknots for the 15 targets, SimRNA and FARFAR2 produce 6 and 2 structures that contain pseudoknots, respectively; however, none of the detected pseudoknots by SimRNA and FARFAR2 have correct correspondence to those in the native structures. DRfold predicts two structures that have pseudoknots assigned, as shown in Figure S2. It can be observed that DRfold can correctly recover the pseudoknots in both cases, highlighting the ability of the DRfold networks to learn complex inter-nucleotide interaction patterns.

Figure S2. Comparison of secondary structures for native and DRfold structure for targets 7o80AT and 7o7zAH, respectively. Red color highlights the assigned pseudoknots.

12. The Reviewer commented:

8) There are some DL-based models to predict RNA 3D structure. I wonder why DRfold has not been compared with these methods... In my opinion, the section on comparative analysis is minimalist. Most of the existing prediction methods have not been taken into account in comparison. No comparison was made with recent approaches based on DL models. Other DL methods for RNA 3D structure prediction:

- FARFAR+ARES (<https://pubmed.ncbi.nlm.nih.gov/34446608/>)

- DeepFoldRNA (<https://doi.org/10.1101/2022.05.15.491755>)

- E2Efold-3D (<https://doi.org/10.48550/arXiv.2207.01586>)

- <https://doi.org/10.1101/2021.08.30.458226>

The methods are available on GitHub.

Thank you for raising this very important suggestion. Following the suggestion, we have added the three publicly available methods in our benchmark experiment, including FARFAR+ARES, DeepFoldRNA and E2Efold-3D (now named RhoFold). Here, we skip the method described in <https://www.biorxiv.org/content/10.1101/2021.08.30.458226> in our test, as we could not find the programs. In addition, we considered two other recently developed deep-learning based methods, RoseTTAFoldRNA and trRosettaRNA. A new section was added to summarize the comparison results of DRfold with these new control methods (Page 10):

DRfold produces competitive predictions to cutting-edge deep learning methods

Most recently, several deep learning models have been proposed for RNA structure prediction. **Table S6** summarized the results of DRfold on the 40 test RNAs compared to five publicly released deep learning methods: ARES¹⁷, DeepFoldRNA²⁴, RhoFold²³, RoseTTAFoldRNA³⁷, and trRosettaRNA²⁵. Depending on the input features that the models were trained on, these methods can be classified into either single sequence-based or multiple sequence alignment (MSA)-based approaches. Although MSA-based methods can benefit from co-evolution information derived from MSAs and therefore often achieve better performance on overall structure prediction^{18, 38}, training on single query sequences has advantages in terms of the speed and flexibility of modeling as the procedure does not rely on the construction of MSAs, which can often be tedious and complicated³⁹.

The data in **Table S6** show that DRfold significantly outperforms other single sequence-based approaches, including the previous control methods and FARFA2/ARES, with p-values as determined by Student's t-tests <E-06. Although DRfold was trained on single sequences, it achieved comparable performance with most of the MSA-based approaches. For instance, DRfold had a higher TM-score (0.435) than RhoFold (0.420) and RoseTTAFoldRNA (0.428), but lower than DeepFoldRNA (0.485) and trRosettaRNA (0.474); the differences between them were not statistically significant except for DeepFoldRNA which had a p-value=1.66E-02 against DRfold.

It should be noted that a non-redundancy filter between the training and test datasets was stringently implemented for DRfold, but the same was not implemented between the training sets of the control methods and the 40 test targets used in this study. For example, we found that 26 targets in our test dataset had a sequence identity above 90% to the trRosettaRNA training dataset. If we exclude these 26 targets, the average TM-score of trRosettaRNA will be reduced to 0.422, which is considerably lower than that of DRfold (0.476). This result again suggests that the performance of most deep learning RNA structure prediction methods may depend on the sequence similarities between the target and training sequences.

From a methodological perspective, the aforementioned deep learning methods can be classified as end-to-end approaches (RhoFold and RoseTTAFoldRNA) or geometry-based approaches (trRosettaRNA and DeepFoldRNA). DRfold, however, combines both approaches through potential integration. The integration provides DRfold with flexibility in its pipeline expansion. For instance, if we simply use DeepFoldRNA's geometry predictions to construct the geometry potential for DRfold, without any further parameter optimization, the hybrid pipeline achieves an average TM-score of 0.501, which is 3.3% higher than the best individual program, DeepFoldRNA (0.485), for the 40 test targets (Table S6), highlighting the potential methodological flexibility and expandability of DRfold.

Table S6. Overall performance of different RNA structure prediction methods on 40 test RNAs. Methods are split into two categories depending on whether they are trained on single sequence or multiple sequence alignment (MSA), while the 'Hybrid' at the bottom row refers to the hybrid approach using the geometric restraints of DeepFoldRNA to guide DRfold folding simulations. P-values are two-tailed Student's t-test calculated between DRfold and each individual control methods. The bold fonts highlight the best performing value in each category.

Starting from	Methods	TM-score (p-value)	RMSD (p-value)
Single sequence	3DRNA	0.251 (5.79E-07)	20.53 (7.35E-05)
	SimRNA	0.196 (2.64E-08)	23.88 (6.14E-07)
	BRiQ	0.216 (2.47E-07)	22.88 (3.34E-07)
	FARFAR2	0.203 (4.35E-08)	22.48 (3.72E-07)
	RNAcomposer	0.239 (1.05E-06)	20.80 (1.90E-04)
	FARFAR2+ARES	0.195 (2.53E-08)	22.82 (1.35E-06)
	DRfold	0.435	14.44
MSA	DeepFoldRNA	0.485 (1.66E-02)	12.19 (1.90E-01)
	RhoFold	0.420 (4.40E-01)	11.57 (2.34E-02)
	RoseTTAFoldRNA	0.428 (5.89E-01)	14.61 (8.36E-01)
	trRosettaRNA	0.474 (9.75E-02)	10.94 (8.80E-02)
Hybrid	DRfold/DeepFoldRNA Potential	0.501 (1.66E-05)	10.65 (4.41E-05)

13. The Reviewer commented:

9) In addition to the evaluation based on comparing predictions with the reference structures, was any validation of the predicted models done? I mean, for example, checking clash-score, torsion angles, bond lengths, the handedness of helices, etc... Analysis of computer-generated models (Carrascoza et al., 2022; doi:10.1261/rna.078685.121) shows that prediction methods rarely implement functions to check and correct stereochemical parameters. Do the models built by DRfold have the correct structure parameters?

We thank the Reviewer for raising this important question. In the updated version, DRfold implements a new two-step full-atom refinement procedure, following the L-BFGS structure optimization, which we found improved the stereochemical parameters of the DRfold models.

Accordingly, we first added a new section of “**Atomic-level structure refinement**” to describe the refinement procedure (Page 17):

Atomic-level structure refinement

Since both the end-to-end pipeline and the L-BFGS folding simulations operate on the reduced 3-bead model, we implement a two-step procedure to reconstruct and refine the full atomic model of DRfold. During the first step, we use Arena (<https://zhanggroup.org/Arena/>) to construct the standard conformations of the full-atomic structure for each of the four types of nucleotides, based on the generic A-form RNA helix with a 32.7° twist and a 2.548 \AA rise⁴⁸, which are then superimposed to the three-atom frame (P, C4', N) of the coarse-grained DRfold model to quickly obtain the initial full-atomic RNA structure. In this step, three fast refinement steps are taken to rectify incorrect bond lengths and angles, base and base pair conformations, and atom clashes, respectively, while keeping the input 3-bead model frozen. Next, a full-atom MD minimization is performed using OpenMM³⁵ to further refine the local structure geometry, including steric clash and bond-length/angle violation removal. The MD force field is based on AMBER14³⁶ and specified by 'amber14-all.xml' and 'amber14/tip3pfb.xml' in the package. For each model, $k \times N_{atoms}/20$ minimization steps are run, where $k = 0.6$ is the empirical coefficient and N_{atoms} is the number of atoms in the full-atomic structure.

Next, we added a new section entitled “**Structure refinement improves the physical realism of DRfold models**” to discuss the physical validation and local structural quality of the DRfold models, which show that the structural refinement steps play an important role in improving the local quality/physical realism of the predicted structures (Page 9):

Structure refinement improves the physical realism of DRfold models

The learning models and potentials utilized in DRfold are coarse-grained built on the P, C4', and N atoms (**Figure 1F**). To enhance the validity and biological usefulness of the predicted models, a two-step refinement process was designed to obtain atomic-level, fine-grained models. **Table S5** presents a comparison of the RNA structure validity parameters through different stages of structural refinement. The raw DRfold models exhibit a high clash score (224.15), resulting in a high MolProbity score³⁴ (3.95). Arena, an in-house program (see **Methods**), was employed to quickly reconstruct full-atom RNA structures and rectify incorrect base conformations on top of raw coarse-grained models, termed as Step 1. Subsequently, the clash and MolProbity scores decreased to 82.79 and 3.42, respectively. Step 2 involved the use of OpenMM³⁵ to further refine the structures through AMBER-guided³⁶ all-atom molecular dynamics (MD) minimization, leading to a significant decrease in the clash (18.57) and MolProbity scores (2.42). Accordingly, the bond length and torsion angle violations, assessed by the root mean square (RMS) deviations from their restrained ideal values, were reduced from 0.06 and 10.99 to 0.03 and 4.21, respectively, after the two-step refinement procedure.

The validity parameters from experimental structures in the PDB are also listed as a reference at the bottom of **Table S5**. Prior to refinement, the MolProbity score of DRfold was 63.2% higher than that of native structures. Following refinement, the MolProbity score difference between the DRfold models and experimental structures decreased to 16.9%. Moreover, the clash score, bond length and torsion angle variations of the final DRfold models all became much closer to those of the experimental structures after the atomic level structure refinement. As a tradeoff, the global model quality assessed by TM-score suffered only a very minor decrease from 0.439 to 0.435.

Table S5. Comparison of RNA structure validity parameters at different steps of structural refinement. Clash score is calculated as the number of serious clashes per 1000 atoms, obtained from the MolProbity program. RMS (bond) and RMS (angles) are root mean square deviations of bond lengths and torsion angles of the DRfold models from their restrained ideal values. Refinement Step 1 refers to the application of Arena to construct full-atom models. Refinement Step2 refers to the application of OpenMM MD simulation package to refine the full-atom models. “Experimental” refers to the target structures in the PDB.

	Clash score	RMS (bond)	RMS (angles)	MolProbity score
Raw DRfold	224.15	0.06	10.77	3.95
Refinement Step 1	82.79	0.05	6.59	3.41
Refinement Step 2	18.57	0.03	4.21	2.83
Experimental	7.25	0.01	1.13	2.42

14. The Reviewer commented:

10) There are several coarse-grained models developed for RNA. Why did the authors choose this particular 3-bead model?

We use the 3-bead model (C4', P, N) because these three atoms represent the structural centers of the nucleotide, phosphate backbone, and base which are critical to specify the global structure of the RNA conformations. Meanwhile, the full-atom models can be effectively retrieved from the 3-vector virtual bond system. We have added Figure 1F and the following paragraph to clarify the point (Page 14):

Here, we use a three-bead model, as specified by the C4', P, and glycosidic N atoms of the nucleobase (**Fig. 1F**), to represent the coarse-grained conformation of a nucleotide. These three atoms (C4', P, N) represent the structural centers of the nucleotide, phosphate backbone, and nitrogenous base, respectively, which are critical to determine the global structure of the RNA conformation. The full-atom models can be effectively recovered from the 3-vector virtual bond system.

15. The Reviewer commented:

Minor:

- "For examples, methods" -> "For example, methods"
- "ID sequence" is an awkward phrase, sequence is ID by default, ID structure is an alternative term for the sequence, so use "sequence" or "ID structure"
- "structurewhere, where" -> "structure, where"
- "RNAComposor" -> "RNAComposer"
- "other top methods which requires" -> "other top methods which require"

We are very grateful for the Reviewer's careful reading of our manuscript. We have corrected those and other typos that we found accordingly.

Response to Reviewer #2

We very much appreciate the comments and suggestions from the Reviewer, which helped to significantly improve the quality and description of the manuscript. The major concern from this Reviewer is on the details of the methodology and experimental settings. In the updated version, we have included additional specifics regarding DRfold as well as the setups of the control methods. Especially, we added additional experimental analyses for different sequence identity cutoffs between the test and training dataset to examine the impact of sequence homology thresholds on the DRfold performance. Below, we include point-by-point replies to the comments of the Reviewer, where all changes have been highlighted in yellow in the manuscript.

1. The Reviewer commented:

Inspired by the success of deep learning method in protein 3D structure prediction of protein, its application to that of RNA is expected. DRfold is such a method. It uses similar deep learning framework as Alphafold2 but has some novel features such as using query sequence and predicted secondary structures as input, using coarse-grained end-to-end learning, using two independent transformer blocks to learn local features and geometric restraints of RNA 3D structure simultaneously and converts them into a composite potential to optimize the three-atom coarse-grained structure of RNA. DRfold outperforms the control methods on the test set and also showed good performance in CASP15. DRfold provides a new approach to predict RNA 3D structure and I recommend it for publication after some minor revisions.

We appreciate the nice summary and positive comments of the Reviewer on the work.

2. The Reviewer commented:

1. For the control methods, it should clearly give what their inputs are. For examples, what secondary structures are used by RNAComposer and 3dRNA in the prediction? Predicted by RNAfold or PETFold? What procedure is used in the prediction by 3dRNA? Assemble or Optimize?

We thank the Reviewer for raising these questions. To address the issue, we added a new section (Text S1) in **Supplementary Materials** to introduce the configurations of the control methods:

Text S1. A brief introduction of the configurations of the control methods

Multiple RNA structure prediction methods have been used as control methods in our benchmark tests. Among the traditional approaches, RNAComposer¹, FARFAR2⁵ and 3dRNA² are the representative fragment assembly methods, while RNA-BRiQ³ and SimRNA⁴ are the two representative *ab initio* RNA structure prediction methods.

The predictions of RNAComposer and 3dRNA were directly obtained by feeding their web servers with query sequences and secondary structure predictions from RNAfold⁶. All other options were kept unchanged. More specifically, for 3dRNA, the “_routine” and “_ss_method” parameters were “assemble” and “RNAfold” respectively.

RNA-BRiQ, SimRNA, and FARFAR2 were installed locally and provided with sequence information and predicted secondary structures from RNAfold. The “BRiQ_Predict” command was used to predict RNA structures for RNA-BRiQ. For SimRNA, the “SimRNA” command was first used with the “-E” option set to 10. The “clustering” command was then used for clustering, followed by the “SimRNA_trafl2pdbs” command with the “AA” option to extract final predictions.

For FARFAR2, the “rna_denovo” command was used with default settings and a maximum running time of 72 hours. Final predictions were selected based on the minimal energy.

Additionally, 5 deep learning based methods, including ARES⁷, DeepFoldRNA⁸, RhoFold⁹, RoseTTAFoldRNA¹⁰ and trRosettaRNA¹¹, were also considered for benchmark. All these methods were installed locally with the default settings. Note that ARES was configured to perform the conformation selection from the structures generated by FARFAR2.

3. The Reviewer commented:

2. In DRfold, the secondary structure is predicted by two complementary methods: RNAfold and PETfold. How is the predicted SS matrix obtained from the results both of them? Since PETfold uses MSA as input, does the result of DRfold also depend on MSA?

We thank the Reviewer for pointing out the ambiguousness. Here, consistent with the requirement of DRfold which starts from single sequence, we only used single sequence as the input for PETfold. In the first paragraph of **Feature preparation and embedding**, we have added the following clarification (Page 13):

Here, consistent with the requirement of DRfold, both RNAfold and PETfold are configured with sequence input only.

4. The Reviewer commented:

3. Modified nucleotides can be mutated into standard nucleotides, why are they classified separately? or what does DRfold do with RNAs containing "N"? Is it mutated to a standard base or removed?

During the training, we considered the non-standard nucleotides as a separate state to avoid possible noise brought by the uncertainty of unknown and modified nucleotides. Therefore, our model can treat RNAs with 5 states (‘A’, ‘U’, ‘G’, ‘C’, ‘N’) in the coarse-grained modeling. But at the last stage of full-atomic structure reconstruction and refinement, we will mutate ‘N’ to the smallest base ‘U’ (for unpaired residues) or the conjugated base (for paired residues). In the first paragraph of **Feature preparation and embedding**, we have added the following clarification (Page 12):

The only required input of DRfold is the nucleotide sequence, which is represented by a 5-D one-hot encoded vector, including 4 types of nucleotides (‘A’, ‘U’, ‘G’, ‘C’) and an unknown state (‘N’) representing modified or degenerate nucleotides. The last state is added to avoid possible training noise brought by the uncertainty of the nucleotides. Therefore, the DRfold model can model RNAs with 5 states (‘A’, ‘U’, ‘G’, ‘C’, ‘N’). At the last stage of atomic structure reconstruction and refinement, however, the residue ‘N’ will be mutated to either the smallest base ‘U’ (for unpaired residues) or the conjugated base (for paired residues) for full-length RNA structure prediction.

5. The Reviewer commented:

4. In line 371, how is the predefined conformation specified for each nucleotide?

The following details with citation have been added to the second paragraph **RNA structure module** for clarification (Page 14):

The predefined conformations were obtained by collecting the resulting local structures after performing symmetric orthogonalization⁴² on coordinates of each nucleotide type in the ideal A-form RNA helix structure.

6. The Reviewer commented:

5. The RNAs in the training set are collected from sequence cluster centers (sequence identity cutoff of 90%) of solved structures deposited in or after the year 2021 in the PDB : is the prediction accuracy sensitive to the cutoff? For example, 80%.

This is a very good question. In this manuscript, we performed additional data analysis on four different sequence identity cutoffs from 90% to 60%. While we found that the prediction accuracy of DRfold is indeed sensitive to the sequence homology cutoffs, DRfold consistently outperforms the previous control methods among all cut-off thresholds. We have added the following paragraph to clarify the point (Page 5):

To investigate the possible impact of sequence homology cutoffs on the accuracy of the DRfold models, an exhaustive test considering various sequence identity cut-offs between the training and test sets was conducted. Following conventional criteria used in previous studies for RNA structure/SS prediction using deep learning^{11, 23, 24, 25}, additional datasets were constructed by excluding targets with sequence identities greater than multiple thresholds (i.e., 80%, 70% and 60%) to the DRfold training dataset; this resulted in 32, 23, and 10 sequence-nonredundant RNA structures, respectively. The results show that the performance of DRfold correlates with the sequence cut-offs for the test sets, where the average TM-scores of the selected test targets gradually decreased from 0.435 to 0.309 as the maximum sequence identity cut-off decreased from 90% to 60% (**Figure 2D**). These data suggest that the deep learning-based predictions are more reliable when trained on similar sequences. Nevertheless, the average TM-score for DRfold consistently exceeded that of the best control methods by at least 33.9% across all thresholds.

Figure 2D. The correlation between the TM-score and the sequence identity cut-off.

In addition, we presented the secondary structure pairing results of DRfold models using two different sequence identity cutoffs of 90% and 80% in Table S1, respectively. The results also show that the performance in base-pairing by DRfold slightly decreased when the sequence identity cutoff was reduced. But DRfold consistently outperformed the control methods at both threshold cutoffs. We have added the following paragraph to summarize the results (Page 6):

The hydrogen-bonding interactions between conjugated nucleotides are critical to stabilize the tertiary structures and functions of RNAs. It is therefore useful to investigate whether and how DRfold can recover these SS patterns. In **Table S1**, we summarize the base interaction network fidelity (INF)^{26, 27} and deformation index (DI)²⁸ scores of the models generated by DRfold and the control methods, which were calculated using the RNA-Puzzles toolkit²⁹. Here, the INF was split into four categories, including Watson-Crick (INF_wc), non-Watson-Crick (INF_nwc), stacking (INF_stack), and overall interactions (INF_all). Although DRfold does not employ specific base-pairing related potentials, it outperforms other methods across each evaluation index, suggesting that the relative frame positions in the frame aligned point error (FAPE) and geometrical potentials may have implicitly helped DRfold to recover the base pairing patterns of the structure models (see **Methods**). In the lower panel of **Table S1**, we also list the performance comparisons on the targets with a sequence identity cut-off of 80% to the DRfold training set, where DRfold still showed an advantage compared to the automatic control methods. It is noted that despite the overall advantage, the success rate of non-canonical base pairing prediction (INF_nwc) by DRfold was still low. A more detailed learning model at the atomic level trained on the datasets with enhanced non-canonical pairing samples might help improve the accuracy for INF_nwc.

Table S1. Comparison of RNA base pairing related metrics between DRfold and control methods in the test set with a sequence identity cutoff of 90% and 80% to the training set respectively. The bold fonts highlight the best performing value in each category.

Methods	DI	INF_all	INF_wc	INF_nwc	INF_stack
Sequence identity cutoff <90%					
3dRNA	40.30	0.586	0.633	0.067	0.597
FARFAR2	43.92	0.604	0.589	0.042	0.644
RNAComposer	41.17	0.616	0.638	0.142	0.628
RNA-BRiQ	39.36	0.622	0.613	0.095	0.645
SimRNA	50.52	0.528	0.384	0.012	0.616
DRfold	26.27	0.708	0.768	0.155	0.711
Sequence identity cutoff <80%					
3dRNA	36.68	0.581	0.625	0.046	0.593
FARFAR2	41.72	0.596	0.575	0.033	0.638
RNAComposer	46.76	0.605	0.616	0.122	0.622
RNA-BRiQ	43.41	0.616	0.599	0.069	0.641
SimRNA	40.90	0.525	0.377	0.000	0.612
DRfold	31.63	0.673	0.722	0.126	0.678

Response to Reviewer #3

We very much appreciate the comments and suggestions from the Reviewer, which helped to significantly improve the quality of the manuscript. The major concern from this Reviewer is on the lack of comparison of DRfold with newly developed deep learning-based approaches. In the revised manuscript, we have added new comparison results to 5 start-of-the-art deep learning RNA prediction methods, including ARES, DeepFoldRNA, RhoFold, RoseTTAFoldRNA, trRosettaRNA, for more comprehensive benchmark tests. The Reviewer also requested benchmark tests on using different secondary structure models, where we added reports on the modeling results based on SPOT-RNA and target secondary structure assignments. Finally, we carefully addressed other important concerns from this Reviewer, including redundancy threshold between test and training targets, interpretation of TMscore-length correlation data, and literature citation of representative methods. Below, we include point-by-point replies to the comments of the Reviewer, where all changes have been highlighted in yellow in the manuscript.

1. The Reviewer commented:

This paper demonstrates that ab initio RNA 3D structure prediction can be achieved using end-to-end deep learning similar to AlphaFold2, and that the prediction accuracy is better than that of existing RNA 3D structure prediction methods. Considering that the number of RNA 3D structures available as training data is an order of magnitude smaller than that of proteins, it is very reasonable to adopt a coarse-grained three-atom model instead of a full-atom model like AlphaFold2, and to incorporate the RNA secondary structure predictions. It is unfortunate that even with these innovations, the results of the blind test on CASP15 still fall short of the existing approaches.

We thank the Reviewer for the constructive summary and positive comments on the manuscript.

2. The Reviewer commented:

p.3, l.60: “SPOT-RNA and e2efold”, p.7, l.245: “Previous studies 8,9”: In this paper, these two methods are listed as methods for RNA secondary structure prediction using deep learning. However, e2efold has been shown in later studies to suffer from severe overfitting and is inappropriate to be listed as a representative study; MXfold2[1] and Ufold[2] would be more appropriate.

*[1] Sato et al. 2021. “RNA Secondary Structure Prediction Using Deep Learning with Thermodynamic Integration.” *Nature Communications* 12 (1): 941.*

*[2] Fu et al. 2022. “UFold: Fast and Accurate RNA Secondary Structure Prediction with Deep Learning.” *Nucleic Acids Research* 50 (3): e14.*

We thank the Reviewer for pointing out the issue. We have accordingly modified the citation and statement as the following (Page 3):

Deep machine learning has recently demonstrated promising performance in RNA structure feature prediction. For example, SPOT-RNA¹¹, MXfold2¹² and Ufold¹³ utilize convolutional neural networks (CNNs) or recurrent neural networks (RNNs) to improve the accuracy of secondary structure (SS) prediction for RNAs.

3. The Reviewer commented:

p.4, l.96: “sequence identify cutoff of 90%” Sequences with 90% sequence identity are extremely similar, and with such similar sequences in the training data, it is not surprising how easy it is to predict the structure. Since many benchmarks for RNA secondary structure prediction use a cutoff of 80% sequence identity, it is necessary to set the cutoff at least at the same level. You should also indicate how homologous the 40 sequences in the test data are to the training data, and whether there is a correlation between homology to the training data and the accuracy of the 3D structure prediction.

We thank the Reviewer for raising this important question. In the revised manuscript, we performed additional data analyses on four different sequence identity cutoffs from 90% to 60%. While we found that the prediction accuracy of DRfold is indeed sensitive to the sequence homology cutoffs, DRfold consistently outperformed the previous control methods among all these cut-off thresholds. We have added the following paragraph to clarify the point (Page 5):

To investigate the possible impact of sequence homology cutoffs on the accuracy of the DRfold models, an exhaustive test considering various sequence identity cut-offs between the training and test sets was conducted. Following conventional criteria used in previous studies for RNA structure/SS prediction using deep learning^{11, 23, 24, 25}, additional datasets were constructed by excluding targets with sequence identities greater than multiple thresholds (i.e., 80%, 70% and 60%) to the DRfold training dataset; this resulted in 32, 23, and 10 sequence-nonredundant RNA structures, respectively. The results show that the performance of DRfold correlates with the sequence cut-offs for the test sets, where the average TM-scores of the selected test targets gradually decreased from 0.435 to 0.309 as the maximum sequence identity cut-off decreased from 90% to 60% (**Figure 2D**). These data suggest that the deep learning-based predictions are more reliable when trained on similar sequences. Nevertheless, the average TM-score for DRfold consistently exceeded that of the best control methods by at least 33.9% across all thresholds.

Figure 2D. The correlation between the TM-score and the sequence identity cut-off.

In addition, we presented the secondary structure pairing results of DRfold models using two different sequence identity cutoffs of 90% and 80% in Table S1, respectively. The results also show that the performance in base-pairing by DRfold slightly decreased when the sequence

identity cutoff was reduced. But DRfold consistently outperformed the control methods at both threshold cutoffs. We have added the following paragraph to summarize the results (Page 6):

The hydrogen-bonding interactions between conjugated nucleotides are critical to stabilize the tertiary structures and functions of RNAs. It is therefore useful to investigate whether and how DRfold can recover these SS patterns. In **Table S1**, we summarize the base interaction network fidelity (INF)^{26, 27} and deformation index (DI)²⁸ scores of the models generated by DRfold and the control methods, which were calculated using the RNA-Puzzles toolkit²⁹. Here, the INF was split into four categories, including Watson-Crick (INF_wc), non-Watson-Crick (INF_nwc), stacking (INF_stack), and overall interactions (INF_all). Although DRfold does not employ specific base-pairing related potentials, it outperforms other methods across each evaluation index, suggesting that the relative frame positions in the frame aligned point error (FAPE) and geometrical potentials may have implicitly helped DRfold to recover the base pairing patterns of the structure models (see **Methods**). In the lower panel of **Table S1**, we also list the performance comparisons on the targets with a sequence identity cut-off of 80% to the DRfold training set, where DRfold still showed an advantage compared to the automatic control methods. It is noted that despite the overall advantage, the success rate of non-canonical base pairing prediction (INF_nwc) by DRfold was still low. A more detailed learning model at the atomic level trained on the datasets with enhanced non-canonical pairing samples might help improve the accuracy for INF_nwc.

Table S1. Comparison of RNA base pairing related metrics between DRfold and control methods in the test set with a sequence identity cutoff of 90% and 80% to the training set respectively. The bold fonts highlight the best performing value in each category.

Methods	DI	INF_all	INF_wc	INF_nwc	INF_stack
Sequence identity cutoff <90%					
3dRNA	40.30	0.586	0.633	0.067	0.597
FARFAR2	43.92	0.604	0.589	0.042	0.644
RNAComposer	41.17	0.616	0.638	0.142	0.628
RNA-BRiQ	39.36	0.622	0.613	0.095	0.645
SimRNA	50.52	0.528	0.384	0.012	0.616
DRfold	26.27	0.708	0.768	0.155	0.711
Sequence identity cutoff <80%					
3dRNA	36.68	0.581	0.625	0.046	0.593
FARFAR2	41.72	0.596	0.575	0.033	0.638
RNAComposer	46.76	0.605	0.616	0.122	0.622
RNA-BRiQ	43.41	0.616	0.599	0.069	0.641
SimRNA	40.90	0.525	0.377	0.000	0.612
DRfold	31.63	0.673	0.722	0.126	0.678

4. The Reviewer commented:

p.4, l.100: structurewhere → structure

The typo was corrected. Thank you.

5. The Reviewer commented:

p.4, l.105: “DRfold outperforms current RNA structure predictors.” This section shows that the proposed method has good prediction accuracy compared to existing RNA 3D structure methods without deep learning. However, in recent years, several deep learning-based RNA 3D structure

prediction methods similar to the proposed method have been reported. It is necessary to compare the proposed method with them.

- Feng et al. 2022. “Accurate de Novo Prediction of RNA 3D Structure with Transformer Network.” **bioRxiv**. <https://doi.org/10.1101/2022.10.24.513506>.

- Zhang et al. 2022. “Physics-Aware Graph Neural Network for Accurate RNA 3D Structure Prediction.” **arXiv [cs.LG]**. *arXiv*. <http://arxiv.org/abs/2210.16392>.

- Baek et al. 2022. “Accurate Prediction of Nucleic Acid and Protein- Nucleic Acid Complexes Using RoseTTAFoldNA.” **bioRxiv**. <https://doi.org/10.1101/2022.09.09.507333>.

- Pearce et al 2022. “De Novo RNA Tertiary Structure Prediction at Atomic Resolution Using Geometric Potentials from Deep Learning.” **bioRxiv**. <https://doi.org/10.1101/2022.05.15.491755>.

- Shen et al. 2022. “E2Efold-3D: End-to-End Deep Learning Method for Accurate de Novo RNA 3D Structure Prediction.” **arXiv [q-bio.QM]**. *arXiv*. <http://arxiv.org/abs/2207.01586>.

We are grateful for the important comment. Following the suggestion, we have added four publicly available methods, i.e., trRosettaRNA, RoseTTAFoldRNA, DeepFoldRNA and E2Efold-3D for performance comparison. FARFAR2+ARES is also considered. We skip Zhang et al. 2022. “Physics-Aware Graph Neural Network for Accurate RNA 3D Structure Prediction.” as we could not find the program from the websites. The deep learning methods were tested with two types of input: MSA and single sequence. We find that DRfold, although trained only on single sequence information, produced comparable results to the state-of-the-art deep learning methods based on MSAs. A hybrid approach by integrating DRfold with the geometric restraints from another in-house MSA-based program DeepDoldRNA outperformed all tested programs in our experiment.

Accordingly, we added a new section, entitled “**DRfold produces competitive predictions with third-party deep learning methods**”, to summarize the comparison results of DRfold with these new deep learning-based approaches (Page 10):

DRfold produces competitive predictions to cutting-edge deep learning methods

Most recently, several deep learning models have been proposed for RNA structure prediction. **Table S6** summarized the results of DRfold on the 40 test RNAs compared to five publicly released deep learning methods: ARES¹⁷, DeepFoldRNA²⁴, RhoFold²³, RoseTTAFoldRNA³⁷, and trRosettaRNA²⁵. Depending on the input features that the models were trained on, these methods can be classified into either single sequence-based or multiple sequence alignment (MSA)-based approaches. Although MSA-based methods can benefit from co-evolution information derived from MSAs and therefore often achieve better performance on overall structure prediction^{18, 38}, training on single query sequences has advantages in terms of the speed and flexibility of modeling as the procedure does not rely on the construction of MSAs, which can often be tedious and complicated³⁹.

The data in **Table S6** show that DRfold significantly outperforms other single sequence-based approaches, including the previous control methods and FARFAR2/ARES, with p-values as determined by Student’s t-tests <E-06. Although DRfold was trained on single sequences, it achieved comparable performance with most of the MSA-based approaches. For instance, DRfold had a higher TM-score (0.435) than RhoFold (0.420) and RoseTTAFoldRNA (0.428), but lower than DeepFoldRNA (0.485) and trRosettaRNA (0.474); the differences between them were not statistically significant except for DeepFoldRNA which had a p-value=1.66E-02 against DRfold.

It should be noted that a non-redundancy filter between the training and test datasets was stringently implemented for DRfold, but the same was not implemented between the training sets of the control methods and the 40 test targets used in this study. For example, we found that 26 targets in our test dataset had a sequence identity above 90% to the trRosettaRNA training dataset. If we exclude these 26 targets, the average TM-score of trRosettaRNA will be reduced to 0.422, which is considerably lower than that of DRfold (0.476). This result again suggests that the performance of most deep learning RNA structure prediction methods may depend on the sequence similarities between the target and training sequences.

From a methodological perspective, the aforementioned deep learning methods can be classified as end-to-end approaches (RhoFold and RoseTTAFoldRNA) or geometry-based approaches (trRosettaRNA and DeepFoldRNA). DRfold, however, combines both approaches through potential integration. The integration provides DRfold with flexibility in its pipeline expansion. For instance, if we simply use DeepFoldRNA's geometry predictions to construct the geometry potential for DRfold, without any further parameter optimization, the hybrid pipeline achieves an average TM-score of 0.501, which is 3.3% higher than the best individual program, DeepFoldRNA (0.485), for the 40 test targets (**Table S6**), highlighting the potential methodological flexibility and expandability of DRfold.

Table S6. Overall performance of different RNA structure prediction methods on 40 test RNAs. Methods are split into two categories depending on whether they are trained on single sequence or multiple sequence alignment (MSA), while the 'Hybrid' at the bottom row refers to the hybrid approach using the geometric restraints of DeepFoldRNA to guide DRfold folding simulations. P-values are two-tailed Student's t-test calculated between DRfold and each individual control methods. The bold fonts highlight the best performing value in each category.

Starting from	Methods	TM-score (p-value)	RMSD (p-value)
Single sequence	3DRNA	0.251 (5.79E-07)	20.53 (7.35E-05)
	SimRNA	0.196 (2.64E-08)	23.88 (6.14E-07)
	BRiQ	0.216 (2.47E-07)	22.88 (3.34E-07)
	FARFAR2	0.203 (4.35E-08)	22.48 (3.72E-07)
	RNAcomposer	0.239 (1.05E-06)	20.80 (1.90E-04)
	FARFAR2+ARES	0.195 (2.53E-08)	22.82 (1.35E-06)
	DRfold	0.435	14.44
MSA	DeepFoldRNA	0.485 (1.66E-02)	12.19 (1.90E-01)
	RhoFold	0.420 (4.40E-01)	11.57 (2.34E-02)
	RoseTTAFoldRNA	0.428 (5.89E-01)	14.61 (8.36E-01)
	trRosettaRNA	0.474 (9.75E-02)	10.94 (8.80E-02)
Hybrid	DRfold/DeepFoldRNA Potential	0.501 (1.66E-05)	10.65 (4.41E-05)

6. The Reviewer commented:

p.5, l.150: "Interestingly, after excluding the 5 extreme targets with length > 200, the PCC becomes extraordinarily different, i.e., 0.71 (see dashed line in Figure 2D). Such correlation would suggest that the performance will be better with longer (< 200) RNA targets." The linear regressions in Figure 2D do not even appear to be a good fit because the prediction accuracies are so scattered. A statistical test on the regression coefficient will tell us whether these regressions are a good fit or not. In any case, the above statement is overly optimistic because it is an extrapolation for the portion of the sequence length that exceeds 200.

We thank the Reviewer for pointing out this issue. We have calculated the *P*-value of the regression fit for the points with $L < 200$ Nts and it is reported as $3.82E-07$. However, we agree with the Reviewer that the former statement and conclusion may be misleading as they ignore the points with $L > 200$ NTs. To avoid confusion, we have rewritten the paragraph in which we removed the misleading (or overly optimistic) conclusion and meanwhile discussed the possible reason and solution to the failure of folding large-size RNAs (Page 5-6, please note that the original Fig 2D was moved to Figure S1 in Supplementary Materials):

In **Figure S1**, we list the scatter plot of TM-score versus the length of the test RNAs, where a weak correlation (Pearson Correlation Coefficient, $PCC = -0.20$) can be observed, indicating that the

performance of DRfold is overall weakly dependent on the RNA length. It is notable that for those targets with lengths > 200 NTs, the TM-scores obtained by DRfold are lower overall than those obtained for smaller targets < 200 NTs. One reason for the suboptimal performance for large-size RNAs is probably that a maximum RNA length cutoff was set to 200 NTs when we trained the models in DRfold due to the limited GPU memory (with a single Nvidia A40 GPU with 32 GB memory) used during the training, and therefore the interaction patterns for extremely distant (>200) nucleotide pairs may not be sufficiently learned. Developing length-insensitive variants of attention networks by utilizing more comprehensive RNA dataset and larger computing resources should help DRfold to learn the longer-range inter-residue interactions and therefore enhance its ability to fold large-sized RNA structures.

7. The Reviewer commented:

p.7, l.242: “the significant importance of the SS embedding feature” Given the importance of RNA secondary structure, it would be nice to have a comparison not only with the combination of RNAfold and PETFold, but also with other prediction methods or with the correct secondary structures obtained from the correct 3D structures given.

We thank the Reviewer for the constructive suggestion. Accordingly, we performed a new experiment of DRfold with three different SS features. We found that although DRfold could generate higher TM-score models when utilizing the target SS than that utilizing predicted SS, the difference is quite modest, indicating that DRfold is not overly sensitive to SS features once a reasonable SS model is used, as the MCCs for both predictors were above 0.65 (Table S4). We may need more sophisticated SS information, e.g., (predicted) pseudoknots or interactions between helices, to further leverage the SS feature for high-accuracy RNA structure prediction. We have added the following paragraph to discuss the result (Page 9):

Given the special role of SS in RNA tertiary structure prediction, we further tested DRfold with two other types of SS inputs from either SPOT-RNA predictions or extracted from the target structures. **Table S4** compares the average TM-score and RMSD of raw DRfold predictions under three conditions with different SS features. Compared to the default settings, the SPOT-RNA-based SS feature provides comparable overall performance, with a slightly lower average TM-score but lower average RMSD, despite DRfold not being trained with this SS model. As expected, the SS features extracted from the experimental structures (Ground-Truth in **Table S4**) yield the best performance compared to other SS features. However, the superiority is somewhat limited, as the average TM-score using ground-truth SS features is only 0.91% and 2.78% higher than that using two predicted SS features respectively. This may be partly because DRfold has been trained on the predicted SS information with noises and therefore the modeling weights associated with the SS component are not strong enough to count for the true SS assignments; in other word, the SS weights might be even stronger if the DRfold model was retrained on the target SSs to count for the ground-truth SS assignments. But the overall data suggest that the current DRfold model is not overly sensitive to the SS feature, as long as SS predictions are of reasonable accuracy, where all tested SS models have an MCC above 0.670. On the other hand, the current 2D SS features from traditional Watson-Crick base-pairing predictions are relatively simple, where input matrices with more specific SS information, such as pseudoknots or inter-helical interactions, may be necessary to further leverage the local SS features for more accurate RNA structure prediction.

Table S4. Performance comparison of DRfold with secondary structures predicted by default (consensus of RNAfold and PETFold), SPOT-RNA, and Ground-Truth secondary structure. MCC refers to the Matthews correlation coefficient between predicted and target secondary structure assignments.

SS prediction methods	MCC	TM-score	RMSD
Default	0.678	0.439	14.49
SPOT-RNA	0.727	0.433	13.61
Ground-Truth	1.000	0.443	13.17

8. The Reviewer commented:

p.7, l.252: “Blind RNA structure prediction in CASP15” This section reports the results of blind test on CASP15. The discussion here is based on the z-score of RMSD and TM score for relative accuracy comparison with other methods. However, to determine the generalization ability of the proposed method, a comparison with experiments on the training and test data prepared in this paper based on the raw RMSD and TM score is necessary.

We thank the Reviewer for the good suggestion. In the revised manuscript, we calculated the average TM-score and RMSD for groups in CASP15 that have submissions for all targets in Table S9. First, the relative rank of different groups, as well as the relative rank of DRfold (rDP) to other groups, on RMSD/TM-score were largely consistent with those by Z-score in Tables S7-8. Second, the average TM-score/RMSD on the CASP15 targets ($=0.288/21.60 \text{ \AA}$, or $0.302/20.34 \text{ \AA}$ after excluding the super-long 720 Nts target R1138) was largely consistent with the benchmark results with the 60% sequence identity cutoff ($0.309/24.27 \text{ \AA}$, Figure 2D), demonstrating the robustness of the benchmark test and generalization ability of the DRfold on modeling different RNA structures.

We have added the following paragraph in the Section “**Blind RNA structure prediction in CASP15**” (Page 11):

In Table S9, we further list the comparisons of average RMSD and TM-score of the groups that have submissions for all CASP15 targets, where DRfold ranks 4th and 9th on TM-score and RMSD, respectively, which are largely consistent with its ranking on Z-scores. Meanwhile, we found that the average values of TM-score/RMSD on the CASP15 targets ($=0.288/21.60 \text{ \AA}$, or $=0.302/20.34 \text{ \AA}$ after excluding the super-long 720 NTs target R1138) are largely consistent with the benchmark test results on the targets with a sequence identity cutoff 60% to the training dataset ($=0.309/24.27 \text{ \AA}$). Considering that all CASP15 targets also have a sequence identity below 60% to the DRfold training dataset, this result demonstrates the robustness of the benchmark test and generalization ability of the DRfold on modeling different RNA structures.

Table S9. Group performance of first model for average RMSD and TM-score respectively. Groups that have submitted models for all targets are considered.

Rank	Group ID	RMSD (\AA)	Rank	Group ID	TM-score
1	AIchemy_RNA2	14.03	1	AIchemy_RNA2	0.485
2	Chen	15.48	2	Chen	0.432
3	RNApolis	15.90	3	RNApolis	0.401
4	rDP	21.60	4	Yang-Server	0.305
5	Yang-Server	21.85	5	UltraFold	0.295
6	UltraFold	23.12	6	CoMMiT-human	0.294
7	UltraFold_Server	23.43	7	CoMMiT-server	0.291
8	CoMMiT-server	23.55	8	Kiharalab	0.291
9	CoMMiT-human	23.72	9	rDP	0.288
10	Kiharalab	24.46	10	UltraFold_Server	0.286
11	Coqualia	25.75	11	SoutheRNA	0.281

12	SoutheRNA	28.15	12	SHT	0.280
13	SHT	28.95	13	GWxraylab	0.276
14	GinobiFold	29.65	14	Coqualia	0.273
15	FoldEver	31.20	15	GinobiFold	0.270
16	GWxraylab	31.61	16	Manifold	0.246
17	Manifold-E	31.97	17	Manifold-E	0.242
18	Manifold	32.98	18	FoldEver	0.196
19	Graphen_Medical	41.80	19	Graphen_Medical	0.171
20	Kiharalab_Server	82.57	20	Kiharalab_Server	0.164

REVIEWERS' COMMENTS

Reviewer #1 (Remarks to the Author):

I appreciate the additional work the authors did to improve the manuscript and compare their method with other algorithms. The work was carefully done, well summarized, and clearly described. Also, thank you for your honest answer regarding the size of the predicted structures. I accept and acknowledge it. All of my concerns have been addressed in the revision.

There is one additional thing I would like to point out. A month ago, a paper on the new CASP competitions was published, written by the organizers, which would be worth quoting where the authors write about RNA prediction in CASP (see Kryshtafovych et al. "New prediction categories in CASP15", *PROTEINS: Structure, Function, and Bioinformatics* 2023; doi:). Surely the organizers would feel appreciated in this way for their efforts in organizing the new competition.

Reviewer #2 (Remarks to the Author):

The authors have answered all my questions and made the corresponding modifications. I have no further comments and recommend this manuscript for publication.

Reviewer #3 (Remarks to the Author):

Thank you for your mostly appropriate responses to my comments. I have a few additional comments on the revised parts of the paper.

I.203 and I.218: For the reader's convenience, it would be nice to include definitions of interaction network fidelity (INF), deformation index (DI), the mean of circular quantity (MCQ), and handedness score in the paper.

I.318: "<E-06" → The value of the mantissa part is missing.

I.318: DFfold → DRfold

I.400: "use DeepFoldRNA's geometry predictions to construct geometry potential for DRfold ..."
I do not understand what you are doing with just this description. Please be more specific. Also, this combination has the best accuracy compared to other methods, including DRfold alone. It would be very useful to discuss why this combination was not made the main result of this paper and why this combination is more accurate.

I.326: "DRfold not being trained with this SS model"

Does this mean that the model trained with the secondary structures from RNAfold and PETfold is used as is, but DRfold is not trained with the secondary structures predicted by SPOT-RNA or the correct secondary structures? Since neither RNAfold nor PETfold can predict pseudoknotted structures, it would be a waste to apply the same model to the SPOT-RNA predicted and correct structures. If possible, please compare the results with retraining using SPOT-RNA predicted and correct structures. In addition, please include the results without the use of secondary structures in Table S4.

Response to Reviewer #1

We very much appreciate the comments and suggestions from the Reviewer, which we found very helpful for improving the quality of the DRfold manuscript. In the revision, we have added the suggested citations accordingly. In the following, we include point-by-point replies to the comments of the Reviewer, where all changes have been highlighted in yellow in the manuscript.

1. The Reviewer commented:

I appreciate the additional work the authors did to improve the manuscript and compare their method with other algorithms. The work was carefully done, well summarized, and clearly described. Also, thank you for your honest answer regarding the size of the predicted structures. I accept and acknowledge it. All of my concerns have been addressed in the revision.

We appreciate the positive comments on our previous revision.

2. The Reviewer commented:

*There is one additional thing I would like to point out. A month ago, a paper on the new CASP competitions was published, written by the organizers, which would be worth quoting where the authors write about RNA prediction in CASP (see Kryshchuk et al. "New prediction categories in CASP15", *PROTEINS: Structure, Function, and Bioinformatics* 2023; doi:). Surely the organizers would feel appreciated in this way for their efforts in organizing the new competition.*

Thank you for the good suggestion. We have added the suggested citation as Ref [40] when introducing the performance of DRfold in CASP15 in Section “**Blind RNA structure prediction in CASP15**” (Page 11):

An early version of the automated DRfold program participated in the recent community wide CASP15 experiment for RNA structure prediction⁴⁰ with Group ID ‘rDP’.

40. Kryshchuk A, et al. New prediction categories in CASP15. *Proteins: Structure, Function, and Bioinformatics*, in press (2023) doi: 10.1002/prot.26515.

Response to Reviewer #2

1. The Reviewer commented:

The authors have answered all my questions and made the corresponding modifications. I have no further comments and recommend this manuscript for publication.

We are glad that the Reviewer is satisfied with our Revision.

Response to Reviewer #3

We very much appreciate the comments and suggestions from the Reviewer, which help to further improve the quality of the manuscript. In the updated version, we have included additional discussions to explain several unclear concepts and rationales pointed out by the Reviewer. Meanwhile, following the Reviewer's suggestion, we performed additional experiments to examine the impact of model retraining on the final performance of DRfold. Below, we include point-by-point replies to the comments of the Reviewer, where all changes have been highlighted in yellow in the manuscript.

1. The Reviewer commented:

Thank you for your mostly appropriate responses to my comments. I have a few additional comments on the revised parts of the paper.

We thank the Reviewer for accepting most of our responses.

2. The Reviewer commented:

l.203 and l.218: For the reader's convenience, it would be nice to include definitions of interaction network fidelity (INF), deformation index (DI), the mean of circular quantity (MCQ), and handedness score in the paper.

We appreciate the good suggestion by the Reviewer. Accordingly, we have added definitions of interaction network fidelity (INF) and deformation index (DI) in Section “**DRfold outperforms previous RNA structure predictors**” (Page 6):

Here, the INF is defined as Matthews Correlation Coefficient (MCC) between the interactions of the reference structure and that of the predicted structure, and it was split into four categories according to the interaction types, including Watson-Crick (INF_wc), non-Watson-Crick (INF_nwc), stacking (INF_stack), and overall interactions (INF_all). The DI is defined as the RMSD between two optimally aligned 3D structures divided by the base INF and can reflect the overall features (encoded by the RMSD) calibrated by the quality of the reproduced interaction network (encoded by the INF value).

MCQ and Handedness score are two metrics for local structure comparisons, which are defined now in Section “**DRfold outperforms previous RNA structure predictors**” (Page 6):

Here, the MCQ score measures the dissimilarity between two structures in torsion angle space using full-atom representations³⁰, assuming the standard bond lengths and bond angles are constant values. In addition, the Handedness score is introduced to evaluate the correctness of the chirality of the RNA helices, which computes the fraction rate of non-loop residues in the predicted models that have closer C4' torsion angles to the targets than to the mirror images of the target structures.

3. The Reviewer commented:

l.318: "<E-06" → The value of the mantissa part is missing.

We thank the Reviewer for pointing out the possible misleading description. We have modified it into $\leq 1.05E-06$ (Page 10):

The data in **Table S7** show that DRfold significantly outperforms other single sequence-based approaches, including the previous control methods and FARAFA2/ARES, with p-values as determined by Student's t-tests $\leq 1.05E-06$.

4. The Reviewer commented:

l.318: DFfold → DRfold

Thank you for pointing out the typo which is now fixed.

5. The Reviewer commented:

l.400: “use DeepFoldRNA’s geometry predictions to construct geometry potential for DRfold ...” I do not understand what you are doing with just this description. Please be more specific. Also, this combination has the best accuracy compared to other methods, including DRfold alone. It would be very useful to discuss why this combination was not made the main result of this paper and why this combination is more accurate.

We thank the Reviewer for the constructive comments, which raised several important issues. First, we have added the following paragraph to describe in more detail the procedure of how we combined DeepFoldRNA with DRfold (Page 11):

From a methodological perspective, the aforementioned deep learning methods can be classified as end-to-end approaches (RhoFold and RoseTTAFoldRNA) or geometry-based approaches (trRosettaRNA and DeepFoldRNA). DRfold, however, combines both approaches through potential integration. The integration provides DRfold with flexibility in its pipeline expansion. For instance, we can use DeepFoldRNA’s geometry predictions to construct hybrid geometry potentials and replace the default geometry potential in DRfold. For this, we combined the end-to-end potentials (from DRfold) and geometry potentials (constructed from DeepFoldRNA predictions) into a new hybrid potential and use it to guide the subsequent structure optimization, while keeping other part of the DRfold procedure unchanged.

Second, we added the following paragraph to discuss why the hybrid approach creates better predictions than the individual pipelines (Page 11):

In such setup, because DeepFoldRNA focuses on training precise inter-nucleotide geometry terms (e.g., distances and orientations) by leveraging coevolution from MSA and unlabeled RNA sequences, it can provide extra and sometime more accurate spatial restraints than the DRfold geometry restraints; thus, a hybrid potential with complementary and more accurate restraints helps better guide the structural assembly and refinement process of DRfold pipeline.

Finally, we added the following paragraph to explain the reason for which this combination was not made the main result of this paper because the major goal of the study was to develop a new standalone pipeline for independent RNA structure prediction (Page 11):

As show in **Table S7**, without any further parameter optimization, the hybrid pipeline (DRfold/DeepFoldRNA) achieves an average TM-score of 0.501, which is 3.3% higher than the best individual program, DeepFoldRNA (0.485), for the 40 test targets. Although the major goal of this study is to develop new standalone pipeline for independent RNA structure modeling, the experiment does show the methodological flexibility of DRfold to combine other methods for further improving its ability for higher-accuracy RNA structure prediction.

6. The Reviewer commented:

1.400: “DRfold not being trained with this SS model” Does this mean that the model trained with the secondary structures from RNAfold and PETfold is used as is, but DRfold is not trained with the secondary structures predicted by SPOT-RNA or the correct secondary structures? Since neither RNAfold nor PETfold can predict pseudoknotted structures, it would be a waste to apply the same model to the SPOT-RNA predicted and correct structures. If possible, please compare the results with retraining using SPOT-RNA predicted and correct structures. In addition, please include the results without the use of secondary structures in Table S4.

We thank the Reviewer for the comments. Yes, "DRfold not being trained with this SS model" means that DRfold was not retrained with secondary structures predicted by SPOT-RNA or the correct secondary structures. To examine the impact of model retraining on the performance, we made an additional experiment, in which we compare the default with retrained models when using new SS assignments, with result summarized in the new Table S5 in SI. It was shown that there is no obvious difference between the default and retrained models, suggesting that default DRfold model is robust and no retraining is needed when using new SS assignments. We have added the following paragraph to discuss the result (Page 9):

One possible reason for the modest improvement from using the native SS assignment is that the DRfold was trained based on predicted SS information with noises and therefore the modeling weights associated with the SS component may not be strong enough to count for the true SS assignments. To examine this possibility, we made a further test on the end-to-end component of the DRfold pipeline with both default and retrained parameters when using new SS assignments. As shown in **Table S5**, although the use of native SS still results in significantly better performance than the predicted SSs, there is no appreciable difference on the average TM-score and RMSD between the models using default and retrained parameters, suggesting that the default models are robust and no retraining is needed when using different SS assignments.

Table S5. Performance comparison of single end-to-end component of the DRfold pipeline with secondary structures predicted by default (consensus of RNAfold and PETfold), SPOT-RNA, and Ground-Truth secondary structure. For SPOT-RNA and Ground-Truth feature, we also report the results based on the models retrained by the corresponding features.

Testing SS feature	TM-score	RMSD (Å)
Default	0.405	13.89
SPOT-RNA	0.405	14.23
SPOT-RNA (Retrained)	0.404	13.67
Ground-Truth	0.423	13.02
Ground-Truth (Retrained)	0.426	12.70

Following the Reviewer’s suggestion, we updated Table S4 with the results without using secondary structures:

Table S4. Performance comparison of DRfold without secondary structure feature, and with secondary structures predicted by default (consensus of RNAfold and PETFold), SPOT-RNA, and Ground-Truth secondary structure. MCC refers to the Matthews correlation coefficient between the predicted and target secondary structure assignments.

SS prediction methods	MCC	TM-score	RMSD (Å)
Without SS	-	0.295	21.10
Default	0.678	0.439	14.49
SPOT-RNA	0.727	0.433	13.61
Ground-Truth	1.000	0.443	13.17